# Ultra-stable and highly reactive colloidal gold nanoparticle catalysts protected using multi-dentate metal oxide nanoclusters

Kang Xia [1], Takafumi Yatabe [1], Kentaro Yonesato[1], Soichi Kikkawa[2], Seiji Yamazoe [2], Ayako Nakata [3], Ryo Ishikawa [4], Naoya Shibata [4], Yuichi Ikuhara [4], Kazuya Yamaguchi [1] & Kosuke Suzuki [1] ✉

Owing to their remarkable properties, gold nanoparticles are applied in diverse fields, including catalysis, electronics, energy conversion and sensors. However, for catalytic applications of colloidal gold nanoparticles, the trade-off between their reactivity and stability is a significant concern. Here we report a universal approach for preparing stable and reactive colloidal small (~3 nm) gold nanoparticles by using multi-dentate polyoxometalates as protecting agents in non-polar solvents. These nanoparticles exhibit exceptional stability even under conditions of high concentration, long-term storage, heating and addition of bases. Moreover, they display excellent catalytic performance in various oxidation reactions of organic substrates using molecular oxygen as the sole oxidant. Our findings highlight the ability of inorganic multi-dentate ligands with structural stability and robust steric and electronic effects to confer stability and reactivity upon gold nanoparticles. This approach can be extended to prepare metal nanoparticles other than gold, enabling the design of novel nanomaterials with promising applications.

Gold nanoparticles have been the subject of extensive investigation in recent decades because of their exceptional reactivity and broad applicability in various research fields, including catalysis, energy conversion, medicine, electronics, optics, magnetic materials and sensors[1–8]. Owing to their large surface area and abundant surface active sites, small gold nanoparticles exhibit high reactivity, which renders them particularly attractive for catalytic applications[1–6]. In this context, intense research attention has been devoted to the development of supported gold nanoparticles to address their inherent instability in past decades; however, the complexity in determining the reactive sites and the difficulty in realising fine-tuning catalysis are critical issues that appear during their applications[1–4]. Hence, exploring gold nanoparticles without supports, which are often called colloidal gold nanoparticles, has become an important avenue towards achieving distinct performance in fine-tuning catalysis[2,3,5,9]. To stabilise

reactive gold nanoparticles in solutions, various chemicals have been employed as protecting agents, such as citric acid, alkanethiols, amines, surfactants and organic polymers (Supplementary Table 1, Entries 1–12)[9–17]. In particular, the introduction of the use of alkanethiols constituted an important breakthrough in the development of small (1–3 nm) gold nanoparticles with long-term (months) stability based on the strong affinity of alkanethiols to gold nanoparticles (Fig. 1a)[10,11]. However, the stability of these gold nanoparticles comes at the expense of their reactivity due to reserved surface active sites and high ligand packing density, which hinder substrate access[9,12,13]. Meanwhile, although organic polymers such as polyvinylpyrrolidone (PVP) are used to prepare gold nanoparticles that perform in alcohol oxidation reactions, their applicability in catalytic reactions requires further validation (Fig. 1a)[17]. Another concern is that organic protecting agents can react under catalytic conditions, undergoing structural

[1]Department of Applied Chemistry, School of Engineering, The University of Tokyo, Tokyo, Japan. [2]Department of Chemistry, Graduate School of Science, Tokyo Metropolitan University, Tokyo, Japan. [3]Research Center for Materials Nanoarchitectonics (MANA), National Institute for Materials Science (NIMS), Ibaraki, Japan. [4]Institute of Engineering Innovation, The University of Tokyo, Tokyo, Japan. ✉e-mail: ksuzuki@appchem.t.u-tokyo.ac.jp

**a** Representative protection methods for gold nanoparticles

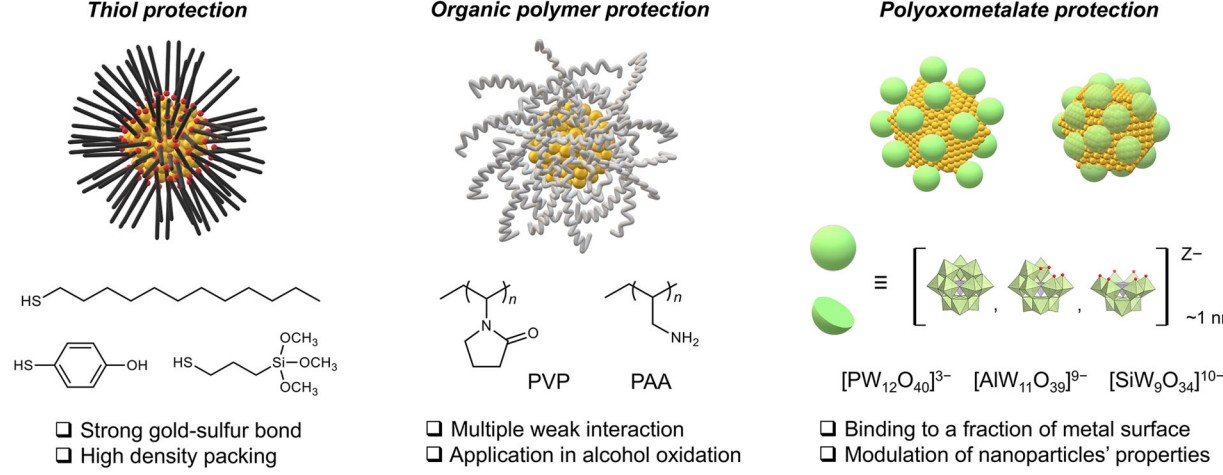

**b** This work: a non-polar-solvent-based multi-dentate polyoxometalate protection method

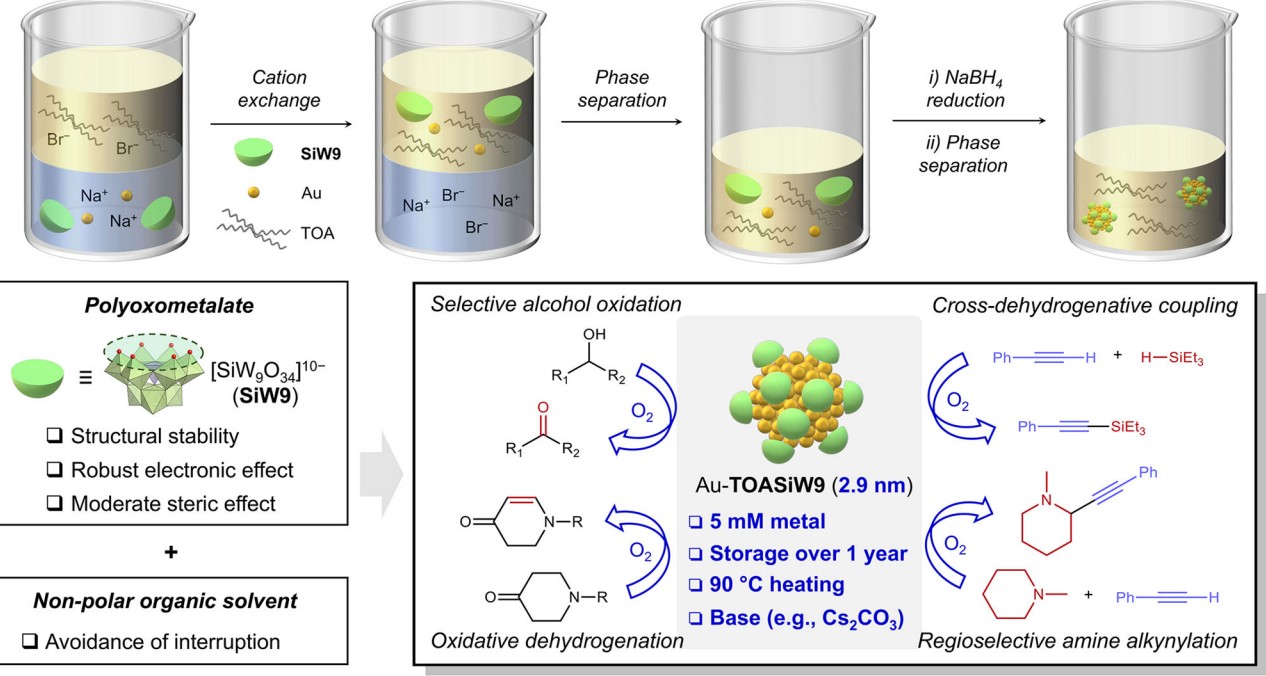

✓ **A universal protocol for achieving gold nanoparticles with high stability, activity and selectivity**
✓ **Wide applicability to various polyoxometalates, surfactants, solvents and metals**

**Fig. 1 | Preparation of gold nanoparticles. a** Representative methods for the preparation of gold nanoparticles using thiol protection, organic polymer protection and polyoxometalate protection. **b** This work: a non-polar-solvent-based multi-dentate polyoxometalate protection method for developing ultra-stable and highly reactive gold nanoparticle catalysts (TOA tetraoctylammonium).

changes and/or detaching from gold nanoparticles during use, leading to destabilisation[12–16]. Therefore, the development of a universal methodology for obtaining stable colloidal gold nanoparticles while maintaining high reactivity for various catalytic reactions is an important yet challenging task.

Metal oxide supports have occupied a central role in the field of metal nanoparticle catalysts owing to their ability to enhance the stability of gold and other metal nanoparticles, control their electronic states and achieve synergistic effects[3,4,6,18]. In this regard, polyoxometalates (POMs), a type of anionic metal oxide nanoclusters with diverse properties and functionalities[19–26], have emerged as efficient protecting agents for stabilising metal nanoparticles by means of

efficient coordination, electrostatic repulsion and steric hindrance (Fig. 1a and Supplementary Table 1, Entries 13 – 24)[27–37]. In addition to an exceptional steric effect, owing to which bulky POM ligands only bind to a fraction of the metal surface, thus ensuring substrate access, the adaptable structures and properties of functional POM ligands enable a molecular-level catalyst design for achieving fine-tuning catalysis and synergistic effects[33–36]. However, POM-protected gold nanoparticles sometimes undergo agglomeration in solution during storage and/or usage[31,32,36,37], which can be attributed to decomposition and structure transformation of POMs in the commonly used aqueous media or occupation of the vacant sites of POMs by alkali metal cations and solvent molecules, leading to destabilization of gold

nanoparticles[38–41]. Moreover, the restriction to hydrophilic use besides the semi-stability issue of POM-protected gold nanoparticles has also limited the exploration of such a feasible molecular-level catalyst design towards practical catalytic applications[32]. Accordingly, we recently applied utilisation of POMs to design various metal oxo clusters and metal clusters in organic solvents, where we have shown that the above-mentioned troubles in aqueous media can be avoided[42–44].

Here we present a feasible method for obtaining ultra-stable and highly reactive small gold nanoparticles (of ~3 nm) by employing multidentate POM ligands in a non-polar solvent system (e.g. toluene, Fig. 1b). Notably, the resultant small gold nanoparticles exhibit exceptional high stability in solution even under harsh conditions such as high concentration (>5 mM metal), long-term storing (>1 year), heating (~90 °C) and addition of bases (e.g. $K_2CO_3$ and $Cs_2CO_3$), which are typically required in catalytic applications. These POM-stabilised gold nanoparticles exhibit high catalytic performance in the aerobic oxidation of alcohols with wide substrate scope and high selectivity to aldehyde or ketone products without changes in the particle size. Additionally, these colloidal gold nanoparticles are effective for various catalytic oxidation reactions using molecular oxygen ($O_2$) as the sole oxidant. This methodology can be extended to various POM ligands, metals (e.g. platinum, ruthenium, rhenium and rhodium) and solvent systems (e.g. p-xylene and 1,2-dichloroethane) to produce small metal nanoparticles stabilised by multi-dentate POM ligands, which demonstrates its wide applicability and versatility.

## Results and discussion
### Development of a non-polar-solvent-based multi-dentate POM protection method

Gold nanoparticles protected by multi-dentate $[SiW_9O_{34}]^{10−}$ (**SiW9**) ligands in toluene (Au-**TOASiW9**) were prepared according to the following method (Fig. 1b): first, separate aqueous solutions of chloroauric acid ($HAuCl_4$) and the sodium salt of **SiW9** (Na**SiW9**) were prepared and transferred into toluene using tetraoctylammonium bromide (TOAB) as a phase transfer agent. Afterwards, an aqueous solution of sodium borohydride ($NaBH_4$) was slowly added to the resulting solution leading to a fast colour change in the toluene phase from orange to dark red (Supplementary Fig. 1), followed by a phase separation process to yield toluene solution of Au-**TOASiW9**. A characteristic surface plasmon resonance band attributed to the gold nanoparticles of Au-**TOASiW9** was observed at 524 nm in the ultraviolet–visible (UV–vis) spectrum (Supplementary Fig. 2). It should be noted that an excess amount of TOAB and a minimal amount of $NaBH_4$ are required; the former facilitates the transfer of **SiW9** as ligands into the toluene phase until reaching the cation exchange equilibrium (Supplementary Fig. 3) and the latter prevents the reverse transfer of **TOASiW9** into the aqueous phase, in which 10 equivalents of TOAB and 4 equivalents of $NaBH_4$ relative to $HAuCl_4$ were found to be the optimal conditions (Supplementary Table 2). Transmission electron microscopy (TEM) observation of Au-**TOASiW9** indicated an average particle size (2.9 nm) comparable to that of dodecanethiol-protected gold nanoparticles (Au-dodecanethiol, 2.6 nm), which were synthesised via the classic Brust–Schiffrin method for comparative purposes (Fig. 2a, e)[10]. Additionally, gold nanoparticles protected only by the surfactant TOAB (Au-TOAB, 2.5 nm) and fully occupied POM $[SiW_{12}O_{40}]^{4−}$ (Au-**TOASiW12**, 3.2 nm) were also prepared (Fig. 2g, h).

Generally, colloidal gold nanoparticles suffer from stability issues such as concentration limitation due to the salting-out effect and agglomeration during use[2,5,9,12]. In contrast, this methodology allows employing metal precursors at concentrations exceeding 5 mM, which are significantly higher than those used in previously reported methods (Supplementary Table 1, Entries 13 − 25). The stability of the small (~3 nm) gold nanoparticles prepared in this study was evaluated under long-term storage, heating treatment and addition of bases, which are

general requirements for practical catalytic applications in solution[1–6,12–15]. Au-**TOASiW9** exhibited exceptional stability, retaining its particle size and size distribution during storage in toluene at room temperature (~25 °C) for over 1 year (Fig. 2b). In contrast, Au-TOAB underwent agglomeration within 2 months (Supplementary Fig. 4). Additionally, Au-**TOASiW9** displayed an extraordinary stability under heating conditions, maintaining its particle size after continuous stirring at 90 °C for 24 h, whereas the particle size of Au-dodecanethiol increased, and Au-TOAB and Au-**TOASiW12** agglomerated and precipitated under the same conditions (Fig. 2c, f–h). Au-**TOASiW9** also showed strong resistance to the addition of bases such as $Cs_2CO_3$, whereas Au-TOAB and Au-dodecanethiol agglomerated and partially precipitated under the same conditions (Supplementary Fig. 5). The precipitation of Au-dodecanethiol can be ascribed to the oxidation of thiol ligands to disulfides followed by detachment from the gold nanoparticles[14], which highlights the stability issue of the ligands themselves during use. Notably, even after participating in catalytic aerobic alcohol oxidations in the presence of a base, Au-**TOASiW9** retained its particle size and size distribution (see Fig. 2d and subsequent discussion). Taken together, these results indicate that Au-**TOASiW9** is much more stable than the well-known Au-dodecanethiol and exhibits resistance to heating and addition of bases, which are essential conditions in catalysis, and multi-dentate POM ligands are key for achieving high stability.

To demonstrate the applicability and versatility of this practical approach, it was successfully expanded to other organic solvents, such as p-xylene and 1,2-dichloroethane, and to various types of POMs, including trivacant **SiW9**, fully occupied **SiW12**, monovacant $[SiW_{11}O_{39}]^{8−}$ (**SiW11**) and divacant $[SiW_{10}O_{36}]^{8−}$ (**SiW10**), furnishing small (~3 nm) gold nanoparticles under similar synthetic conditions (Supplementary Fig. 6). Furthermore, small nanoparticles (<5 nm) of different metals, including platinum, ruthenium, rhenium and rhodium were successfully synthesised (Supplementary Fig. 7).

Next, zeta potential measurements were performed to investigate the surface state of the gold nanoparticles (Fig. 3a). The observed negative zeta potential for the POM-protected gold nanoparticles indicated the formation of anionic POM layers surrounding the gold nanoparticles[31–33]. Au-**TOASiW9** exhibited the most negative zeta potential, suggesting the strongest interparticle electrostatic repulsion that effectively prevents agglomeration. The aberration-corrected annular dark-field scanning TEM (ADF-STEM) images of Au-**TOASiW9** showed the presence of shells on the surface of the gold nanoparticles (Fig. 3b, Supplementary Fig. 8), and elemental mapping using energy-dispersive X-ray spectroscopy (EDS) further confirmed the formation of an Au-core−POM-shell-like structure (Fig. 3c). Through titration experiments of dodecanethiol to Au-**TOASiW9** with inspiration by previous reports[29,30], it was estimated that around 30 **SiW9** ligands surrounded a 3 nm gold particle, and surface coverage can be estimated as 47% (Supplementary Fig. 9, see explanation in detail). Subsequently, X-ray photoemission spectroscopy (XPS) showed that the binding energy of the Au $4f_{7/2}$ region of Au-**TOASiW9** (82.3 eV) was notably lower than that of bulk Au (84.0 eV) and Au-TOAB without POM protection (83.2 eV, Fig. 3d, Supplementary Fig. 10), indicating the anionic status of the POM-protected gold nanoparticles stemming from the electronic interaction between anionic POM ligands and gold nanoparticles[31–36]. Notably, the binding energy of Au-**TOASiW9** (82.3 eV) was even lower than those of reported anionic gold nanoparticles, PVP-protected gold nanoparticles (Au:PVP, 82.7 eV) and **SiW11**-protected gold nanoparticles (82.8 eV), which were obtained in aqueous solution[17,33]. No obvious difference in the binding energies was found between Au-**TOASiW12** and Au-TOAB, highlighting the importance of a robust electronic interaction between multi-dentate POM ligands and gold nanoparticles in this system. Moreover, POM-protected gold nanoparticles with similar particle sizes possessed sequentially modulated electronic states (Fig. 3d), providing a feasible

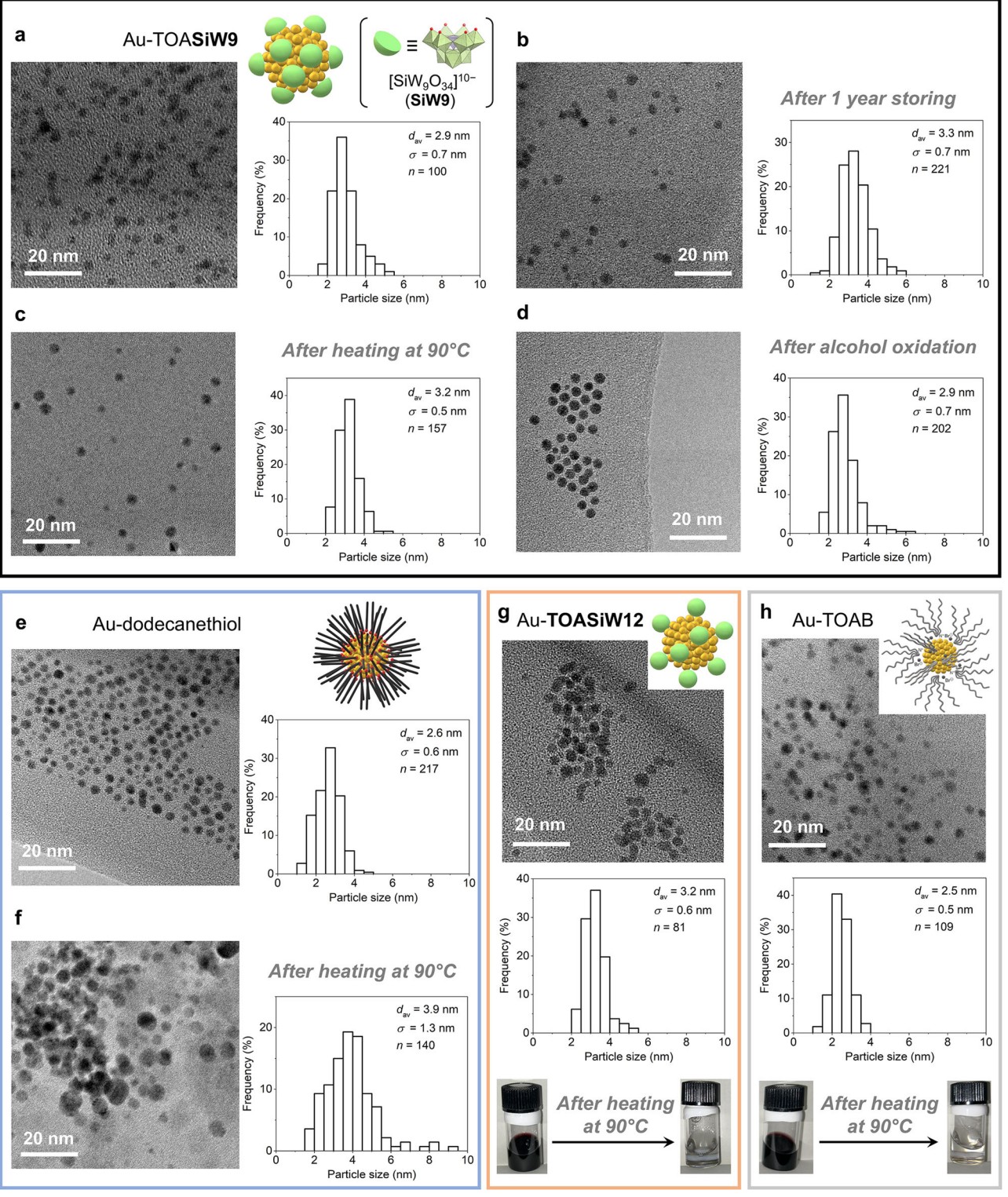

**Fig. 2 | Characterisation and stability test of gold nanoparticles.** TEM images and size distribution histograms: **a** Au-**TOASiW9**. **b** Au-**TOASiW9** after storing for 1 year. **c** Au-**TOASiW9** after heating at 90 °C for 24 h. **d** Au-**TOASiW9** after alcohol oxidation in the presence of Cs₂CO₃ under the conditions shown in Table 1, entry 2. **e** Au-dodecanethiol. **f** Au-dodecanethiol after heating at 90 °C for 24 h. **g** Au-**TOASiW12** (the bottom picture shows the formation of precipitates from Au-**TOASiW12** after heating at 90 °C for 24 h). **h** Au-TOAB (the bottom picture shows the formation of precipitates from Au-TOAB after heating at 90 °C for 24 h).

and effective tool for adjusting the activity of gold nanoparticle catalysts, as previously discussed in detail[36].

To confirm the structures of the POM ligands after hybridisation with gold nanoparticles, solid samples of **TOASiW9** and Au-**TOASiW9** obtained by evaporating the toluene solvent were characterised. Infrared (IR) spectroscopies showed that Au-**TOASiW9** exhibited similar bands to those of **TOASiW9** and **NaSiW9** regarding characteristic peaks in the region from 800 to 1000 cm⁻¹, but differed from those of sodium tungstate, indicating that the structure of **SiW9** was preserved (Supplementary Fig. 11)[34,36]. According to Raman spectroscopies, the

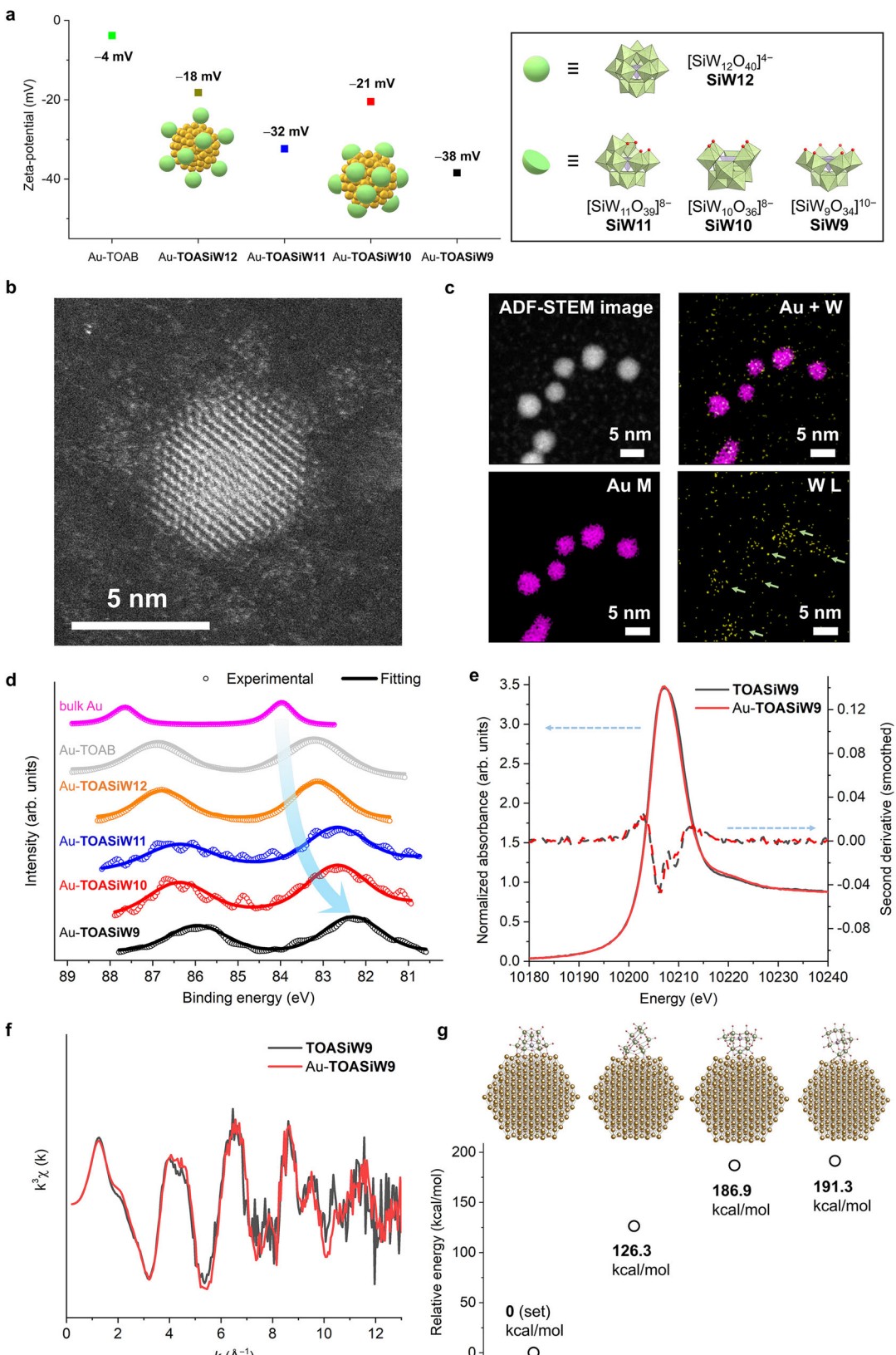

**Fig. 3 | Investigations of the structure and electronic state of Au-TOASiW9.** **a** Zeta potential of Au-**TOASiW9**, Au-**TOASiW10**, Au-**TOASiW11**, Au-**TOASiW12**, Au-TOAB and Au-dodecanethiol in toluene. **b** Representative atomic-resolution annular dark field scanning transmission electron microscopy (ADF-STEM) image of Au-**TOASiW9**. **c** EDS elemental mapping images of Au-**TOASiW9**. **d** XPS spectra of various ligand-protected gold nanoparticles and bulk Au (the arrow indicates the trend of changes in binding energy). **e** W L$_3$-edge XANES spectra with associated second derivatives of **TOASiW9** and Au-**TOASiW9**. **f** $k^3$-Weighted W L$_3$-edge EXAFS spectra of **TOASiW9** and Au-**TOASiW9**. **g**, Relative energy of **SiW9**-protected gold nanoparticles and models of the optimised structures for different orientations of **SiW9**.

characteristic peak of W=O$_d$ bonding from POM structures was observed at 965 cm$^{-1}$ for Au-**TOASiW9** and **TOASiW9**, similar to that of the tetrabutylammonium salt of **SiW9** (TBA$_4$H$_6$SiW$_9$O$_{34}$, **TBASiW9**) at 970 cm$^{-1}$ but slightly shifted from that of the sodium salt of **SiW9** (Na$_{10}$SiW$_9$O$_{34}$, **NaSiW9**) at 940 cm$^{-1}$ (Supplementary Fig. 12). This can be ascribed to the presence of sodium cations near the POM anions in the case of **NaSiW9** increasing the W=O bonding length and weakening the bonding strength[45]. To analyse the precise POM structure, X-ray absorption fine structure analyses were conducted. Similar patterns of Au-**TOASiW9** and **TOASiW9** in the second derivative of white-line region and W L$_3$-edge $k$-space EXAFS spectra, indicating that POMs maintained their structures after hybridization with gold nanoparticles (Fig. 3e, f)[36,46]. Then, similar patterns of **TOASiW9** and **NaSiW9** but completely different from those of the potassium salt of **SiW12** (K$_4$SiW$_{12}$O$_{40}$, **KSiW12**) and WO$_3$ in the second derivative of white-line region indicated that there existed no obvious structural changes in the {WO$_6$} octahedra (Supplementary Fig. 13a, b). The W L$_3$-edge $k$-space EXAFS spectra showed no significant changes between **TOASiW9** and **NaSiW9** while in contrast to that of **KSiW12** and WO$_3$, strongly indicating that POM maintained intact structures as well (Supplementary Fig. 13c). In the $R$-space EXAFS spectra, the peaks at $R$ = 1.2, 1.7 and 3.2 Å assignable to terminal W=O, bridging W−O−W and W−W, respectively, exhibited no drastic changes from **NaSiW9** to **TOASiW9**, further supporting the preservation of POM structures in this method (Supplementary Fig. 13d). Finally, structures of POMs during synthesis of gold nanoparticles were confirmed through deliberately transferring them into aqueous phase (Supplementary Fig. 14, Supplementary Table 2, Entry 5). In the IR spectra, the characteristic peaks of **SiW9** in the region of 500–1000 cm$^{-1}$ were well consistent between **NaSiW9** and POMs after mixing with gold precursors and sodium borohydride respectively, indicating their intact structures in current method (Supplementary Fig. 14). These results demonstrate that POM ligands remain stable in this synthetic system, effectively protecting the metal nanoparticles.

The interaction between **SiW9** and gold nanoparticles was further investigated by performing first-principles calculations. Considering that the interaction of bulky tetraoctylammonium (TOA) cations and toluene solvent molecules with POMs was negligible, structural optimisations were conducted using one multi-dentate **SiW9** ligand with several different orientations on the surface of 2-nm-large gold nanoparticle without TOA cations and solvent molecules (Fig. 3g). The most stable orientation was identified as that in which multi-dentate **SiW9** interacted with a gold nanoparticle at the vacant site of **SiW9**. These results demonstrate that the coordination of gold nanoparticles at the vacant sites of **SiW9** effectively contributes to the nanoparticle stabilisation and, together with the experimental results, highlighting the essential role of an adequate coordination of POM ligands in protecting metal nanoparticles during synthesis, storage and use. Hence, the development of ultra-stable small metal nanoparticles via POM protection was successfully achieved in a non-polar solvent for the first time (Fig. 1b).

## Oxidation reactions catalysed by ultra-stable small gold nanoparticles

The high concentration and extraordinary stability of the developed colloidal gold nanoparticles even under heating conditions and in the presence of bases prompted us to explore their performance in liquid-phase catalytic reactions. Anionic gold nanoparticles have been proved effective in activating O$_2$, which renders them potential catalysts in aerobic oxidation reactions; however, only a few studies have been conducted so far[17,33,36]. Therefore, the low binding energy of the Au 4$f_{7/2}$ region of Au-**TOASiW9** motivated us to investigate the catalytic performance for various aerobic oxidation reactions in solution, particularly those requiring heating treatment or addition of bases, starting with aerobic alcohol oxidation as a typical model reaction of gold nanoparticle catalysts.

Among the examined POM-protected gold nanoparticle catalysts, Au-**TOASiW9** exhibited the highest activity towards the aerobic oxidation of benzyl alcohol (**1a**) in the presence of K$_2$CO$_3$ as a base to selectively furnish benzaldehyde (**2a**) as a product (Table 1 and Supplementary Table 3). A sequentially modulated activity was found depending on the type of POM in line with the electronic states of gold nanoparticles, which can be correlated with the effective activation of O$_2$ on anionic gold nanoparticles (Table 1, Entries 1–5; Fig. 3d and Supplementary Fig. 15). In contrast, gold nanoparticles including Au-**TOASiW12**, Au-TOAB and Au-dodecanethiol exhibited significantly low catalytic activity (Fig. 1a; Table 1, Entries 5–7). Notably, no significant

## Table 1 | Selective aerobic oxidation of benzyl alcohol (1a) to benzaldehyde (2a)[a]

Colloidal gold nanoparticles (Au: 4 mol%)

O$_2$ (1 atm), K$_2$CO$_3$

Toluene, ~25 °C, 24 h

**1a** → **2a**

| Entry | Catalyst | Yield (%) | Particle size (nm) | |
|---|---|---|---|---|
| | | | Before use | After use |
| 1 | Au-**TOASiW9** | 75 | 2.9 | 2.8 |
| 2 | Au-**TOASiW9**[b] | 92 | 2.9 | 2.9 |
| 3 | Au-**TOASiW10** | 17 | 3.4 | 3.3 |
| 4 | Au-**TOASiW11** | 24 | 2.9 | 3.0 |
| 5 | Au-**TOASiW12** | 4 | 3.2 | 4.3 |
| 6 | Au-TOAB | 3 | 2.5 | 5.6 |
| 7 | Au-dodecanethiol | <1 | 2.6 | 2.7 |
| 8 | Au-dodecanethiol[b] | <1 | 2.6 | Precipitation[c] |
| 9 | **TOASiW9** | <1 | – | – |
| 10 | Au-**TOASiW9**, Ar[d] | <1 | 2.9 | – |

[a]Reaction conditions: **1a** (0.25 mmol), 3 mL toluene solution of colloidal gold nanoparticles (Au: 4 mol%), K$_2$CO$_3$ (0.5 mmol), room temperature (~25 °C), O$_2$ (1 atm), 24 h.
[b]Cs$_2$CO$_3$ (0.5 mmol) was used instead of K$_2$CO$_3$ (0.5 mmol).
[c]Disulfide was detected in the reaction solution after the reaction.
[d]Under Ar (1 atm). All the reaction yields were determined via GC analysis using biphenyl as an internal standard.

changes were observed in the particle size of the gold nanoparticles protected with **SiW9**, **SiW10** and **SiW11** even after the reaction, and no significant decrease was observed in the catalytic activity of Au-**TOASiW9** during the reaction (Supplementary Figs. 16 and 17). Furthermore, Raman spectra showed that Au-**TOASiW9** after the catalytic oxidation of **1a** exhibited similar characteristic bands with Au-**TOASiW9**, indicating that the structure of **SiW9** was preserved (Supplementary Fig. 12). In contrast, an evident agglomeration of gold nanoparticles was observed during the reaction using fully occupied **SiW12** as a protecting ligand (Supplementary Fig 17). These results indicated that the presence of vacant (coordination) sites in the POM ligands is essential for protecting the metal nanoparticles. Nevertheless, Au-TOAB underwent more severe agglomeration than Au-**TOASiW12**, indicating that POM ligands can generally function as stabilising agents in comparison to conventional organic substances[31,32]. Considering that the particle size of Au-dodecanethiol was kept after the reaction (Table 1, Entry 7; Supplementary Fig. 17), its low catalytic activity was likely due to difficulties in substrate access and strong bonding of thiol ligands to the surface of gold nanoparticles[12,13]. Meanwhile, Au-**TOASiW9** exhibited higher catalytic activity when using $Cs_2CO_3$ as a base than in the presence of $K_2CO_3$, and **2a** was still selectively produced from **1a** while maintaining the small particle size (Table 1, Entry 2; Fig. 2d). In contrast, the reaction barely proceeded using Au-dodecanethiol and $Cs_2CO_3$ due to the oxidation of the thiol ligands, which led to precipitation of gold nanoparticles (Table 1, Entry 8)[14].

Additionally, **TOASiW9** exhibited no activity and Au-**TOASiW9** did not promote the reaction under argon (Ar) atmosphere, confirming gold nanoparticles as the active sites and $O_2$ as the terminal oxidant in this catalysis (Table 1, Entries 9 and 10). The kinetic isotope effect (KIE) was then examined for Au-**TOASiW9**-catalysed oxidation of **1a**. Under $O_2$ atmosphere (1 atm), a much higher reaction rate was observed for **1a** than benzyl-$\alpha,\alpha$-$d_2$ alcohol ($k_H/k_D = 3.2$, Supplementary Fig. 18), indicating that C−H cleavage can be the turnover limiting step. When the reaction was carried out under air atmosphere ($O_2$, 0.2 atm), no significant KIE was observed ($k_H/k_D = 1.2$, Supplementary Fig. 19), suggesting that $O_2$ adsorption and/or activation can be turnover limiting step under air atmosphere (Supplementary Fig. 15).

To investigate the effect of cations, preparation of gold nanoparticles protected with **SiW9** was examined using different cations such as tetrahexylammonium (THA), tetradecylammonium (TDA), and cetyltrimethylammonium (CTA) instead of TOA. Although TDA can be used for preparation of gold nanoparticles, incomplete phase transfer was observed for THA and CTA owing to less hydrophobicity of metal precursors and emulsion formation, respectively (Supplementary Fig. 20a). The obtained gold nanoparticles using TDA cations and **SiW9** (Au-**TDASiW9**) possessed similar particle sizes of 3 nm and similar catalytic reactivity to those of Au-**TOASiW9**, suggesting that countercations did not necessarily facilitate the reaction (Supplementary Fig. 20b, c). These results provide direct evidence that multi-dentate POM ligands do not only contribute to stabilising the small (~3 nm) gold nanoparticles during their preparation but also allow retaining their catalytically active sites, enabling the modulation of the electronic states of gold nanoparticles for activity control.

Furthermore, the Au-**TOASiW9**-catalysed reaction demonstrated a broad substrate scope, enabling the conversion of various primary and secondary alcohols to the corresponding aldehyde and ketone products, respectively (Fig. 4a, Supplementary Fig. 21). When $Cs_2CO_3$ was used as a base at room temperature (~25 °C), Au-**TOASiW9** efficiently promoted the oxidation of benzyl alcohol as well as benzylic alcohols with either electron-withdrawing or electron-donating groups, affording the corresponding benzaldehydes (**2a**–**2i**). Au-**TOASiW9** successfully catalysed the oxidation of a heteroaromatic alcohol to the corresponding aldehyde (**2j**). In addition, an $\alpha,\beta$-unsaturated alcohol afforded the corresponding $\alpha,\beta$-unsaturated aldehyde (**2k**), and aromatic and aliphatic secondary alcohols also gave the corresponding ketones (**2l**–**2q**).

Finally, the applicability of Au-**TOASiW9** towards various oxidation reactions using $O_2$ as the sole oxidant was confirmed (Fig. 4b, Supplementary Fig. 22). In addition to the high reactivity and wide applicability in alcohol oxidation reactions, Au-**TOASiW9** catalysed the oxidative dehydrogenation of *N*-methyl piperidone in toluene in the presence of $Cs_2CO_3$ as a base (Supplementary Fig. 23a). Subsequently, the cross-dehydrogenative coupling (CDC) reaction of a terminal alkyne and a hydrosilane in the presence of $O_2$ as the hydrogen acceptor efficiently proceeded using Au-**TOASiW9** to afford the desired alkynylsilane without formation of hydrosilylation products. In contrast, when the CDC reaction was performed under the same conditions using typical supported gold nanoparticle catalysts, the undesirable hydrosilylation reaction occurred to a certain extent (Supplementary Fig. 23b)[47]. The observed high selectivity to the CDC product when using Au-**TOASiW9** can be attributed to a fast Au−hydride oxidation induced by activated oxygen species on the anionic gold nanoparticles preventing the hydrosilylation reaction. Moreover, in a regioselective alkynylation of a tertiary amine, Au-**TOASiW9** demonstrated comparable reactivity to the reported catalytic system using supported gold nanoparticles in the presence of $ZnBr_2$ (Supplementary Fig. 22c)[48]. These findings demonstrate that this methodology provides a universal protocol for the preparation of colloidal metal nanoparticle catalysts simultaneously exhibiting high reactivity, stability and selectivity that can compete with conventional supported metal nanoparticle catalysts.

In summary, we have developed a non-polar-solvent-based multidentate POM protection strategy for obtaining ultra-stable and catalytically active colloidal gold nanoparticles. These small gold nanoparticles exhibited remarkable tolerance towards high concentration conditions (>5 mM metal), long-term storage (>1 year), heating treatment (>90 °C) and addition of bases (e.g. $Cs_2CO_3$) without undergoing changes in particle size and size distribution. They exhibited a high reactivity and selectivity in various catalytic oxidation reactions using $O_2$ as the sole oxidant, including alcohol oxidation, piperidone dehydrogenation, terminal alkyne–hydrosilane cross-dehydrogenative coupling and tertiary amine alkynylation. The robust electronic and moderate steric effects of multi-dentate POM ligands are considered essential to achieve an extraordinary catalytic performance. The wide applicability to various POM ligands, solvent systems and metal nanoparticles, and the broad reaction scope render this approach highly promising for solving the compromise between reactivity and stability of metal-nanoparticle-based materials in diverse fields including catalysis, biochemistry, photochemistry, coordination chemistry, pharmaceuticals, physiochemistry and materials science.

## Methods

### Instruments and reagents

Gas chromatography (GC) analyses were conducted on Shimadzu GC2014 equipped with a flame ionization detector (FID) and an InertCap-5 capillary column (30 m × 0.25 mm × 0.25 µm) using Shimadzu CR8A Chromatopac Data Processor for area calculations. GC mass spectrometry (GC-MS) analyses were performed by Shimadzu GCMS-QP2020 equipped with an InertCap-5 MS/NP capillary column (30 m × 0.25 mm × 0.25 µm) at an ionization voltage of 70 eV. Inductively coupled plasma atomic emission spectroscopy (ICP-AES) analyses were conducted by Shimadzu ICPS-8100. Transmission electron microscope (TEM) observations were conducted by JEM-2000EX and JEM-2010F at an acceleration voltage of 200 kV, and scanning transmission electron microscopy (STEM) observations were conducted by JEM-ARM200F Thermal FE at an acceleration voltage of 200 kV and JEM-ARM300CF with a cold FE at an acceleration voltage of 300 kV. The illumination semi-angle and the collection semi-angle for atomic-resolution annular dark field (ADF-)STEM images were acquired 30 mrad and 32–200 mrad, respectively. W $L_3$-edge X-ray absorption spectroscopy (XAS) was carried out at the BL01B1

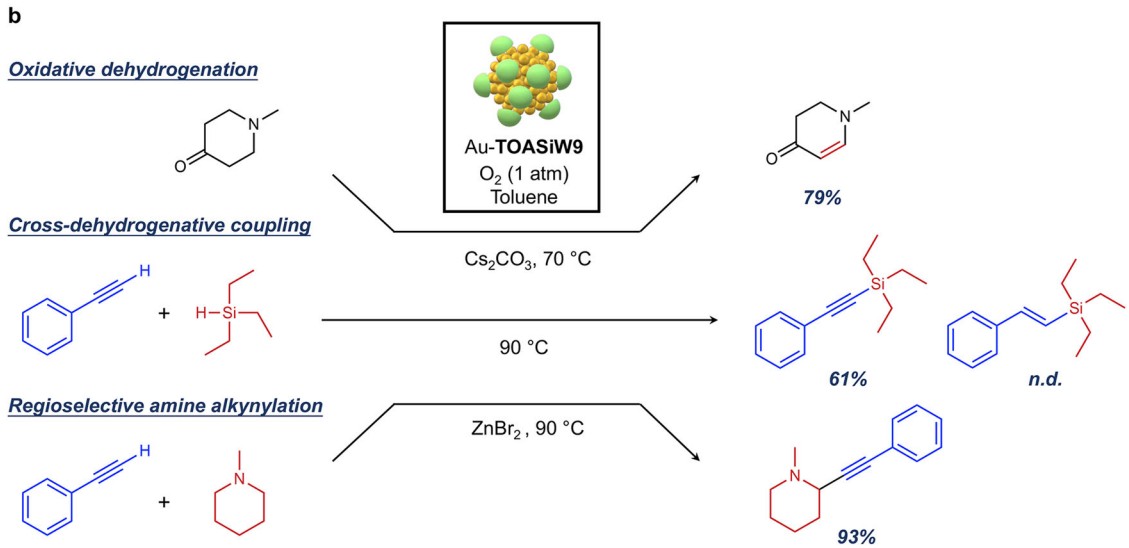

**Fig. 4 | Reaction scope. a** Substrate scope of Au-**TOASiW9**-catalysed aerobic alcohol oxidation. Reaction conditions: alcohol (0.25 mmol), 3 mL toluene solution of Au-**TOASiW9** (Au: 4 mol%), Cs₂CO₃ (0.5 mmol), -25 °C (room temperature), O₂ (1 atm), 24 h. **b** Schematic of various aerobic oxidation reactions catalysed by Au-**TOASiW9**. Detailed reaction conditions are described in Method section.

beamline of SPring-8. X-ray absorption fine structure (XAFS) measurements were conducted in transmission mode using a Si(111) double-crystal monochromator. The X-ray absorption near-edge structure (XANES) and extended X-ray absorption fine structure (EXAFS) spectra were analysed using xTunes programme[49]. Pre-edge backgrounds were subtracted using a McMaster equation. EXAFS backgrounds were subtracted using a cubic spline method (spline range = 5). W L₃-edge EXAFS spectra in $k$-space were obtained as $k^3$-weighted χ spectra after normalization. The X-ray photoelectron spectroscopy (XPS) was performed on ULVAC-PHI PHI5000 Versa-ProbeIII at the Advanced Characterisation Nanotechnology Platform of The University of Tokyo. The samples were embedded in In foil and brought in the introduction chamber. The fitting of experimental data was conducted using a Multipak software (version 9.2.0.5, by Ulvac-phi, inc.) in which the Shirley method was used for the background

and a Gauss−Lorentz type function was performed for fitting. The binding energies were calibrated by using the C 1s signal of C−C bonding at 284.9 eV. Under these conditions, the Au 4$f_{7/2}$ signal of bulk Au (CAS No. 7440-57-5) was located at 84.0 eV. Solution-state ultraviolet−visible (UV−Vis) spectra were measured on JASCO V-770 spectrometer with a 1 cm quartz cell at room temperature (-25 °C). Zeta-potential measurement were conducted on Malvern Zetasizer NanoZS at a backscatter mode and a working voltage of 40 V was adopted. Infra-red (IR) spectra were measured on a JASCO FT/IR-4100 using the attenuated total reflection method. Raman spectra were measured on a JASCO NRS-5100. All chemical reagents were obtained from Tokyo Chemical Industry, Aldrich, Kanto Chemical, or FUJIFILM Wako Pure Chemical (reagent grade) without pretreatment. Inorganic salts of **SiW9**, **SiW10**, **SiW11**, **SiW12** (**NaSiW9**, Na₁₀SiW₉O₃₄; K₈SiW₁₀O₃₆; K₈SiW₁₁O₃₉; K₄SiW₁₂O₄₀) and a tetrabutylammonium salt

of **SiW9** (**TBASiW9**, $(C_{16}H_{36}N)_4H_6SiW_9O_{34}$) were prepared according to the reported procedures[40,50].

## Preparation of gold nanoparticles

Au-**TOASiW9** was prepared as follows: an aqueous solution of $HAuCl_4$ (20 mL, 5 mM) was mixed with a solution of TOAB in toluene (20 mL, 50 mM). The two-phase mixture was vigorously stirred until all the $HAuCl_4$ was transferred into the organic layer, and an aqueous solution of **NaSiW9** (20 mL, 5 mM) was then added to the organic layer and the resulting solution was stirred for 30 min, followed by a phase-separation to yield the organic layer. A freshly prepared aqueous solution of $NaBH_4$ (20 mL, 20 mM) was slowly added, and the organic phase was immediately separated and filtrated twice using hydrophobic filters to remove residual water, affording a toluene solution of Au-**TOASiW9**. For optimization of TOAB and $NaBH_4$ usage (Supplementary Table 2), the amounts of **NaSiW9** and $HAuCl_4$ were hold constant, and the same procedures were used with the exception of varying the amounts of TOAB and $NaBH_4$.

Au-**TOASiW10**, Au-**TOASiW11**, Au-**TOASiW12** and Au-TOAB were prepared using similar procedures as that for Au-**TOASiW9**, except for using different POMs ($K_8SiW_{10}O_{36}$, $K_8SiW_{11}O_{39}$, $K_4SiW_{12}O_{40}$) or no POM in the case of Au-TOAB. Gold nanoparticles in other organic solvents were prepared using the same procedures as described for Au-**TOASiW9** in toluene but using *p*-xylene and 1,2-dichloroethane instead of toluene.

POM-protected platinum, ruthenium and rhenium nanoparticles (Pt-**TOASiW9**, Ru-**TOASiW9**, Re-**TOASiW9**) were prepared as described above for Au-**TOASiW9**, except for using $Na_2PtCl_6$, $K_2RuCl_5$ and $K_2ReCl_6$, respectively, instead of $HAuCl_4$. POM-protected rhodium nanoparticles (Rh-**TOASiW9**) was prepared via a slightly modified methodology using $RhCl_3$ without conducting phase separation until $NaBH_4$ reduction was complete. Au-dodecanethiol was prepared according to the Brust−Schiffrin method[10], except by decreasing the concentration for comparison in catalytic test as follows: an aqueous solution of $HAuCl_4$ (5 mL, 18 mM) was mixed with a solution of TOAB in toluene (20 mL, 20 mM). The two-phase mixture was vigorously stirred until all the $HAuCl_4$ was transferred into the organic layer, and dodecanethiol (0.9 mmol, 10 equivalents to Au) was then added to the organic phase. A freshly prepared aqueous solution of $NaBH_4$ (5 mL, 200 mM) was slowly added with vigorous stirring. After further stirring for 3 h, the organic phase was separated and filtrated twice using hydrophobic filters to remove residual water to give a toluene solution of Au-dodecanethiol, that was directly used without further treatment.

## Titration experiments

UV−vis titration procedure, based on changes in SPR absorbance[29,30], was used to quantify the replacement of **TOASiW9** by thiolates on the surfaces of the gold nanoparticles. A toluene solution of dodecanethiol (5 mM) was gradually added to a toluene solution of Au-**TOASiW9** (0.5 mM Au), and the changes of the absorbance of SPR bands were monitored by UV−vis spectra. The temperature was maintained at $25.0 \pm 0.1\,°C$.

## Procedure for catalytic reactions

The aerobic alcohol oxidation reaction was conducted as follows: **1a** (0.25 mmol), biphenyl as an internal standard (0.25 mmol), Au-**TOASiW9** (2 mL toluene solution containing 0.01 mmol Au), appended toluene for better dispersion of the base (1 mL) and a magnetic stirrer bar were added to a Pyrex glass reactor, which was then purged with $O_2$ gas and sealed with a screw cap. The solution was stirred at room temperature (~25 °C) for 24 h. After completion of the reaction, the substrate conversions and product yields were determined via GC analysis. For the reaction under Ar (1 atm), freeze-pump-thaw cycles were carried out and the reactor was connected to a balloon filled with an Ar gas.

The oxidative dehydrogenation of piperidone was conducted as follows: 1-methyl-4-piperidone (0.25 mmol), biphenyl as an internal standard (0.25 mmol), Au-**TOASiW9** (5 mL toluene solution containing 0.025 mmol Au), and a Teflon-coated magnetic stirrer bar were added to a Pyrex glass reactor, which was purged with $O_2$ gas and then sealed with a screw cap. The solution was stirred at 70 °C for 24 h. After completion of the reaction, the substrate conversions and product yields were determined via GC analysis.

The cross-dehydrogenative coupling reaction of a terminal alkyne and a hydrosilane was conducted as follows: ethynylbenzene (0.25 mmol), triethylsilane (0.3 mmol), biphenyl as an internal standard (0.25 mmol), Au-**TOASiW9** (2 mL toluene solution containing 0.01 mmol Au) and a Teflon-coated magnetic stirrer bar were added to a Pyrex glass reactor, which was purged with $O_2$ gas and sealed with a screw cap. The solution was stirred at 90 °C for 24 h. After completion of the reaction, the substrate conversions and product yields were determined via GC analysis.

The regioselective alkynylation of a tertiary amine was conducted as follows: 1-methylpiperidine (0.5 mmol), ethynylbenzene (0.25 mmol), $ZnBr_2$ (0.25 mmol), biphenyl as an internal standard (0.25 mmol), Au-**TOASiW9** (2 mL toluene solution containing 0.01 mmol Au) and a Teflon-coated magnetic stirrer bar were added to a Pyrex glass reactor, which was then purged with $O_2$ and sealed with a screw cap. The solution was stirred at 90 °C for 24 h. After completion of the reaction, the substrate conversions and product yields were determined via GC analysis.

## Computational methods

First-principles density functional theory calculations were performed by using the CONQUEST code. Double-$\zeta$ plus polarization pseudo atomic orbital (PAO) basis functions were used with norm-conserving pseudopotentials. 5 s and 5p semi-core PAOs were used for Au and W. The PAO ranges [bohr] are as follows: Au [5 s, 5p, 5d, 6 s, 6p] = [2.81, 3.36, (7.12, 3.74), (7.12, 3.74), 7.12], Si [3 s, 3p, 3d] = [(7.12, 4.02), (7.12, 4.02), 7.12], W [5 s, 5p, 6 s, 5d, 6p] = [3.22, 3.85, (7.97, 4.18), (7.97, 4.18), 7.97], and O [2 s, 2p, 3d] = [(4.91, 2.58), (4.91, 2.58), 4.91]. The PBE exchange-correlation functional was used. The geometry of **SiW9**-protected Au nanoparticle (denoted as Au-**SiW9**) with different orientations was optimised, in which Au was modelled as a nanoparticle of diameter about 2 nm (consisting of 309 atoms) in *Oh* symmetry. According to the formal charge of **SiW9**, we set the charge of the unit cell to be −10 for Au-**SiW9** with neutralizing background uniform charge density. The cubic unit cells of 75.6×75.6×75.6 bohr³ were used for all systems. The grid spacing was 0.21 bohr.

## Data availability

The data that support the findings of this study are available from the corresponding author upon request.

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

## Acknowledgements

We gratefully acknowledge the financial support from JST FOREST (JPMJFR213M for K.S., JPMJFR2033 for R.I.), JST PRESTO (JPMJPR18T7 for K.S., JPMJPR19T9 for S.Y., JPMJPR20T4 for A.N., JPMJPR227A for T.Y.), JSPS KAKENHI (22H04971 for K.Ya), and the JSPS Core-to-Core programme. XAFS measurements were conducted at SPring-8 with the approval of the Japan Synchrotron Radiation Research Institute (proposal numbers: 2023A1732, 2023A1554, 2022B1860, 2022B1684). A part of this work was supported by Advanced Research Infrastructure for Materials and Nanotechnology in Japan (ARIM) of the Ministry of Education, Culture, Sports, Science and Technology (MEXT), Grant Number JPMXP1222UT0184 and JPMXP1223UT0029. We thank Ms. Mari Morita (The University of Tokyo) for assistance with the STEM-EDS analysis.

## Author contributions

K.S. conceived and directed the project. K.X. designed and performed most of experiments including synthesis, analysis and catalytic reactions. K.Ya. and T.Y. contributed to catalytic reactions and the project. S.K., S.Y. and K.Yo. carried out XAFS measurements and analysis. A.N. performed first-principles calculations. R.I., N.S. and Y.I. contributed to microscopy analysis. K.X. and K.S. wrote the manuscript, with input from all the co-authors.

## Competing interests

The authors declare no competing interests.
