## [Peer Review File · Nature Communications]

Ultra-stable and highly reactive colloidal gold nanoparticle catalysts protected using multi-dentate metal oxide nanoclustersREVIEWER COMMENTS

Reviewer #1 (Remarks to the Author):

The authors report an original and general route to access small metal nanoparticles using a smart synthetic route based on biphasic mixing/extraction/reduction. The synthetic concept is based on earlier referenced works where it was shown that polyoxometalates (POMs) can stabilize metal nanoparticles during formation. Here, the authors utilize a multidentate, highly charged POM which shows enhanced capabilities to form and stabilize highly robust, catalytically active Au NPs. The compounds catalyze a range of industrially important oxidation reactions under mild conditions, using only O₂ as terminal oxidant, which will make this study highly interesting for possible technological further development. This is a major step forward in the field of POM-stabilized metal NPs and has implications for nanomaterials design, catalysis and functional alloy nanostructures. The manuscript is therefore in principle suitable for publication in Nature Communications, once the following (relatively minor) points have been addressed:

1. Figure 1, can the authors clarify what they mean by (i) "binding to a fraction of metal" and (ii) adjustable electronic/basic effect"? I feel that without reading the manuscript, these points are not clear, and my understanding is that the Figure is meant to give a summary of the main properties of each method without having to refer to the manuscript. Also, I suggest to change "relatively large particles sizes....." simply to "particle sizes > 5 nm". In fact, these are still small particles for the NP community!
2. Figure 2: all samples behave as expected, the only notable difference is that in Figure (d), the NPs seem to adopt a hexagonal 2D packing. Is this an artefact of the TEM measurement and of that specific region of the sample, or is this something observed regularly for these samples? This is not relevant to the manuscript as such, but it might be pointing at some interesting POM-mediated interactions
3. POM-cation interactions often significantly affect the supramolecular chemistry of these systems (see reference 37 also). Do the authors see a role of the TOA cations in the effects observed? This could be clarified in the manuscript.

I have also assessed the Supporting Information, which are of high quality and support the findings of the manuscript. In addition, the main manuscript and SI provide all details for reproducing the results. All experimental description are of high quality and clearly written. The data is clearly analyzed and the methodology is sound.

Reviewer #2 (Remarks to the Author):

Suzuki and co-workers from Au(0) nanoparticles in toluene by borohydride reduction of HAuCl₄ in the presence of [SiW₉O₃₄](10⁻), (SiW₉) and a large excess of tetra-n-alkylammonium bromide (TOAB). The material is then characterized mostly in after drying by a routine spectroscopic and microscopic methods and used under strongly basic conditions, i.e., after addition of alkali-metal cations salts of carbonate, CO₃²⁻ as aerobic catalysts for the oxidation of benzyl alcohol to benzaldehyde, and for oxidations of related substrates. The authors also characterize the properties of the final product by comparison to synthetic results obtained starting from SiW₁₂O₄₀(4⁻), SiW₁₁O₃₉(8⁻) and SiW₁₀O₃₆(8⁻). The claimed findings are significant, but considerable work is needed to correct clear errors in understanding the stability of the inorganic clusters and in demonstrating the structures and compositions of the clusters present after treatment with borohydride in synthesis and carbonate in catalysis. Moreover, some key experimental data critical to interpreting the results are missing from the Methods section of the text as well as from the supplementary information. Without that information (specified below), some of the reported results cannot be properly evaluation. Regarding the catalytic studies, additional control experiments, as well as kinetic and analytical data are needed

to provide sufficient evidence for claimed activity and stability. Overall, the above items need to be fully addressed in order to qualify the work for publication in any quality journal of chemistry, inorganic chemistry or catalysis.

An additional problematic issue relates to scholarship and context, i.e., representation of the work in the context of closely related previously-published studies; those very relevant studies are entirely neglected. For example, the closely related work of Richard Finke who prepared organic-solvent soluble catalysts very similar to those reported and proposed here are highly stable and much more reactive than those reported here. However, that entire opus of work, including complexes of Rh and Ir is ignored by the present authors. This is even more problematic in that the present authors propose using POMs in organic solvent to prepare stable colloids of "platinum, ruthenium, rhenium and rhodium", again with no mention of the fact that Finke prepared highly stable and reactive complexes of Rh and Ir over 25 years ago, and published numerous detailed high-quality articles on the "electrosteric stabilization" he observed. For example, the authors refer to the fact that POMs "have emerged as efficient protecting agents for stabilising metal nanoparticles by means of efficient coordination, electrostatic repulsion and steric hindrance". The latter was discovered by Finke but not a single article from his 15 years of work on this topic is cited by the authors. Similarly, Ira Weinstock published numerous articles on stabilization of gold nanoparticles using mono-defect Keggin POM anions. Like Finke's POM-stabilized NPs prepared in organic solvents, Weinstock's are stable for years without aggregation in water and include examples of very small particles, 2-3 nm. Yet, in Figure 1a, the authors summarize "polyoxometalate protection" by showing stabilization of Au NPs by plenary Keggin anions (as opposed to the mono-defect ones used in the works noted above) and then claim that they "agglomerate during storage" and are limited to "relatively large particle sizes". This reference to only plenary POMs, which are quite rare, if reported at all, as the context for their current findings is disturbingly misleading. Finally, Ronny Neuman has published numerous articles on various different metal NPs stabilized in both water and organic solvent by $PV_2Mo_{10}O_{40}(5-)$, and has shown many of those to be highly stable and remarkably reactive and selective catalysts for aerobic oxidations under mild conditions. In addition to all the above, Pavel Kulesza has used phase transfer methods to prepare POM stabilized metal NPs in organic solvents. Not a single citation of any article from Kulesza's opus of work on that topic is cited by the present authors. The aerobic catalysis based on dioxygen activation by POM stabilized Au NPs reported here is also not new, as similar O_2 activation by POM-stabilized gold NPs was reported a few years ago by Wang and co-workers (ref. 29). In the context of the above works, published by several leading groups over many years, the conceptually new finding reported here is effectively disclosure of a synthetic method, demonstrated for a single type of metal nanoparticle (gold). This would be abundantly apparent if through proper scholarly presentation that context were properly included in Introduction and Discussion of the present article. In the context of the above-discussed studies, the new synthetic method reported here and used for a single example (Au) does not justify publication in Nature Commun.

In addition to the above-noted issues of scholarly presentation, there are many points of science that need to be addressed before this work is perhaps submitted elsewhere. These are enumerated below.

I. The stability of SiW9 on Au after synthesis and after catalysis

During the synthesis, the authors combined $HAuCl_4$ with SiW9 and then added $NaBH_4$. Both operations can decompose the POM. It is also well-known that SiW9 will break down in the presence excess Na_2CO_3 . Therefore, it is likely that a large amount of SiW9 decomposed during the catalytic applications.

To address the above issues, the authors should give adequate proof for the stability and presence of pure SiW9 both after synthesis and after catalysis. Notably, no analytical data whatsoever in support of POM stability after catalysis is provided. The authors only measure particle size by TEM! The following experiments are clearly needed.

To address these points the authors should:

1. Report the ESI-MS spectrum of the colloidal solution to verify if other POMs are formed. The IR in

Figure S10 actually indicates that a large fraction of SiW9 decomposed.

2. The authors should purify the colloidal solution using either by dialysis to remove excess POM, or by ultracentrifugation-redispersion to obtain a pure POM-stabilized AuNPs. Check the activity and stability of the purified sample.

3. The authors can readily provide evidence for that POM ligands are bound to gold as shown in their graphics. For this, they simply need to titrate their solutions by addition of thiol determine whether they reach a definitive endpoint. That will also provide quantitative data concerning how many POMs are bound to each gold particle, on average.

4. The data in Supplementary Figure 5 suffer from the omission of key details needed to evaluate the finding. In panel a, the title catalyst, there is a large excess of TOA (10 fold) and of the POM in solution. This is compared with panel c, where small changes are observed for Au stabilized by TOAB. However, nowhere in the text or SI is it indicated the relative amount of TOA present in that sample. The immediate question a reader asks here is why is that data omitted, and if the amount of TOA is less in the sample in panel c, perhaps the slightly lesser stability is attributed to that. The main problem here, again, is one of including all relevant information, with respect to scholarship as noted above and with respect to presentation of all relevant experimental details, as relevant to interpretation of the data in Figure S5.

5. In related work, the authors make a case of the role of the charge of SiW9 by preparing samples using SiW12O40(4-), SiW11O39(8-) and SiW10O36(8-). No data is provided to support the argument that these are intact after synthesis. Sure, the plenary ion is not; it readily converts to the lacunary when not in acidic conditions. The data in Figure 3d are indicated that something is not as assumed by the authors. Namely, The trend in binding energy with charge is not as the large blue arrow indicates. Rather, the binding energy reverses the trend at SiW10. The authors don't comment on this, but it is entirely consistent with some other POM or mixture of POMs present, either for that case or for the SiW12 and SiW11 cases that precede it in the figure. In summary, all new materials prepared and discussed must be fully characterized. In all cases, ESI-MS spectra of the claimed POMs must be provided, along with IR/Raman etc.

In catalysis, the authors report yields but don't comment on whether they checked for products that don't appear in the GC method.

6. As such, the authors should report H-1 NMR spectra of all product mixtures to verify claimed yields in catalysis.

7. TOASiW9W11 should be added to sample reported in entry 6 (Table 1) to confirm the role of the POM in catalysis. The addition can be followed by UV-vis spectroscopy.

8. The authors need to provide kinetic data showing catalyst activity over time, as least for the flagship oxidation of benzyl alcohol. Without this, simply showing the particles remain intact by TEM is far from sufficient: it says nothing about activity and stability of the active surface species during catalysis.

9. As noted earlier, a full set of analytic data, including ESI-MS and vibrational spectroscopy is needed to prove the POM remains stable during catalysis.

10. It might be informative to use PW9 to synthesize AuNPs, and determine the catalytic results. Compare the yields with SiW9-stabilized AuNPs to verify the author's hypothesis that AuNP activity is connected to the XPS results. If this were done, 31-P NMR could also be used to characterize the starting materials and the POM-AuNPs colloidal solutions before and after use.

11. If the dehydrogenation mechanism is correct, O2 should be converted into H2O during the reaction. 18O isotope experiments with 18O2 should be carried out to confirm this.

12. The authors proposed a dehydrogenation mechanism in which the first step is the activation of the substrate to generate hydride. If this is the rate limiting step, a significant kinetic isotope effect should be observed using deuterated benzaldehyde as substrate.

13. In arguing the catalyst is stable under turnover conditions, the authors report (Supporting Figure 5) changes in UV-vis spectra after "addition of Cs2CO3". No indication is given regarding how much Cs2CO3 is added. This is left out entirely. These experiments need to be repeated using the catalytic samples themselves, before and after catalysis. It is very concerning that those obviously necessary data are not included?

14. The catalytic results and stability should be carefully compared to those reported by Finke. This kind of colloid was described by Finke et al. as tetraalkylammonium- and POM-co-stabilized metal(0) NPs. The distinctions and similarities should be stated. The authors should also provide evidence that TOA+ is not involved in the stabilization of AuNPs in this work.

Reviewer #3 (Remarks to the Author):

The paper reports the synthesis of very stable and highly dispersed colloidal Au particles in polar media using Polyoxometalates (POMs) as stabilising ligands. The particles are stable in the presence of a base, and heating which makes them attractive for oxidation reactions typically carried out on Au. The paper carries out systematic experiments and demonstrates well the performance of these materials – however, what is not so clear is the justification of “The robust electronic and moderate steric effects of multi-dentate POM ligands are essential to achieve an extraordinary catalytic performance”. Despite this being a justification for the study (to make stable colloidal particles to study the inherent activity of Au particles), the authors do not separate the catalysts' performance from the stabilizer ligands' role. POMs are known to be highly active in redox especially WO₆ type species reactions – do they contribute to oxygen activation? Are these catalysts more “active” or just more stable so can do more turnovers? Some comparisons to supported samples would help to understand this. In addition, the authors correlate the performance to the nature of the POM suggesting that the POMs which are negatively charged bind to the Au particles and generate negative Au surfaces (via XPS binding energy correlations) – in this case what are the counterions to the Au surfaces? Do the authors suggest electron transport from the POM to the Au?

<As for Reviewer 1>

Comment

The authors report an original and general route to access small metal nanoparticles using a smart synthetic route based on biphasic mixing/extraction/reduction. The synthetic concept is based on earlier referenced works where it was shown that polyoxometalates (POMs) can stabilize metal nanoparticles during formation. Here, the authors utilize a multidentate, highly charged POM which shows enhanced capabilities to form and stabilize highly robust, catalytically active Au NPs. The compounds catalyze a range of industrially important oxidation reactions under mild conditions, using only O₂ as terminal oxidant, which will make this study highly interesting for possible technological further development. This is a major step forward in the field of POM-stabilized metal NPs and has implications for nanomaterials design, catalysis and functional alloy nanostructures. The manuscript is therefore in principle suitable for publication in Nature Communications, once the following (relatively minor) points have been addressed.

Response

Thank you very much for high evaluation of our manuscript and valuable comments. We have carefully considered all your comments and revised the manuscript. We believe that the addition incorporated in the revised manuscript is appropriate. Please confirm the following responses.

Comment

1. Figure 1, can the authors clarify what they mean by (i) "binding to a fraction of metal" and (ii) adjustable electronic/basic effect"? I feel that without reading the manuscript, these points are not clear, and my understanding is that the Figure is meant to give a summary of the main properties of each method without having to refer to the manuscript.

Also, I suggest to change "relatively large particles sizes....." simply to "particle sizes > 5 nm". In fact, these are still small particles for the NP community!

Response

(i) As for the meaning of "binding to a fraction of metal":

Conventional organic ligands usually coordinate to the surface of metal nanoparticles as much as possible to decrease surface free energy, resulting in high ligand packing density and deactivation of surface metals. In contrast, it has been reported that the bulky POMs (diameter ca. 1 nm) bind directly to a fraction of surface metals of the nanoparticle, leaving reactive metals exposed (*Nat. Commun.* **2018**, *9*, 4896). Furthermore, considering high negative charges of POMs, electrostatic repulsion of POMs can not only prevent metal nanoparticles from aggregating but also preserve space between neighboring POMs.

(ii) As for the meaning of "adjustable electronic/basic effect":

The "adjustable electronic/basic effect" is unique properties of using POMs as functional ligands for metal nanoparticles. For example, we have reported that by changing types and anion charges of POMs, the electronic states of gold nanoparticles can be sequentially modulated (*Angew. Chem. Int. Ed.* **2022**, *61*, e202205873). In addition, the basicity of POMs can be used to provide dual functional catalysis in POM-modified gold nanoparticles (*Chem. Commun.* **2022**, *58*, 9018). However, as you pointed out, the meaning of the term is ambiguous, and thus we changed it to "Modulation of nanoparticles' properties".

Based on the above-mentioned points, we revised Fig. 1 as follows:

Fig. 1 | Preparation of gold nanoparticles. **a**, Representative methods for the preparation of gold nanoparticles using thiol protection, organic polymer protection and polyoxometalate protection. **b**, This work: a non-polar-solvent-based multi-dentate polyoxometalate protection method for developing ultra-stable and highly reactive gold nanoparticle catalysts.

Considering the previous papers of preparing small gold nanoparticles of less than 5 nm (*Electroanalysis* **2007**, *19*, 2103–2109; *Nat. Nanotechnol.* **2017**, *12*, 170–176; *Angew. Chem. Int. Ed.* **2017**, *56*, 7083–7087), current subject in the field of POM-protected metal nanoparticles was refined as “semi-stability during storage and/or usage & lack of catalytic application”. Accordingly, the relevant contents were revised as follows:

“However, POM-protected gold nanoparticles sometimes undergo agglomeration in solution during storage and/or usage,^{29,30,34,35} which can be attributed to decomposition and structure transformation of POMs in the commonly used aqueous media or occupation of the vacant sites of POMs by alkali metal cations and solvent molecules, leading to destabilization of gold nanoparticles.³⁶⁻³⁹ Moreover, the restriction to hydrophilic use besides the semi-stability issue of POM-protected gold nanoparticles has also limited the exploration of such a feasible molecular-level catalyst design towards practical catalytic applications.³⁰”

(Page 2, Line 50)

Comment

2. Figure 2: all samples behave as expected, the only notable difference is that in Figure (d), the NPs seem to adopt a hexagonal 2D packing. Is this an artefact of the TEM measurement and of that specific region of the sample, or is this something observed regularly for these samples? This is not relevant to the manuscript as such, but it might be pointing at some interesting POM-mediated interactions.

Response

Thank you for your comment. There exist similar hexagonal 2D packing in Fig. 2d (Au-TOASiW9) and Fig. 2e (Au-dodecanethiol). This may occur during dryness of the sample on the TEM grid, and we think that this is not due to the POM-mediated specific interactions.

Comment

3. POM-cation interactions often significantly affect the supramolecular chemistry of these systems (see reference 37 also). Do the authors see a role of the TOA cations in the effects observed? This could be clarified in the manuscript.

Response

According to your comment, we examined the preparation of POM-modified nanoparticles using several types of tetraalkylammoniums as counter cations. In the present preparation method of POM-modified gold nanoparticles in a nonpolar organic solvent (toluene), we firstly dissolve alkali metal salts of POMs in aqueous solution, followed by transferring POMs into toluene phase as tetraoctylammonium (TOA, C8) salts by using tetraoctylammonium bromide as a phase transfer agent (Supplementary Fig. 20a). Thus, the choice of counter cations is important for this phase transfer process. For example, tetraethylammonium (TEA, C2) and tetrabutylammonium (TBA, C4) salts of POMs cannot dissolve in toluene, and it is difficult to transfer POMs into the toluene phase by using these cations. Use of tetrahexylammonium (THA, C6) as a counter cation resulted in an incomplete transfer of POMs into the toluene phase and a precipitation of gold nanoparticles. In contrast, tetradecylammonium (TDA, C10) salts of POMs can be transferred into the toluene phase, similar to the result of TOA salts. When cetyltrimethylammonium (CTA, C16) was used, toluene and water phase became miscible owing to the emulsion formation, making it difficult to conduct the phase transfer process (Supplementary Fig. 20a). Therefore, we can use TOA and TDA and other potential cations in this method.

As for the particle sizes and catalytic activities, we observed no noticeable difference by changing the cations of POM-modified gold nanoparticles (Supplementary Fig. 20b,c). Therefore, we can conclude that TOA and TDA cations do not directly participate in the catalysis of gold nanoparticles, but are essential in endowing hydrophobicity of inorganic precursors and assisting the stabilization of hybrid nanoparticles.

a Photographs of reaction solutions during phase transfer process

THA (C6):
incomplete transfer

TOA (C8):
successful transfer

TDA (C10):
successful transfer

CTA (C16):
without separation

b TEM results of Au-TDASiW9

c Catalytic results

Catalyst	Yield (%)
Au-TOASiW9	92
Au-TDASiW9	90
TOASiW9	<1
TDASiW9	<1

Reaction conditions: **1a** (0.25 mmol), 3 mL toluene solution of gold nanoparticles (Au: 4 mol%), Cs₂CO₃ (0.5 mmol), room temperature (~25 °C), O₂ (1 atm), 24 h.

Supplementary Fig. 20 | Investigation on the effect of cations during phase transfer process of chloroauric acid into toluene phase with different surfactants. **a**, Photographs of reaction solutions during phase transfer process (top phase, toluene; bottom phase, H₂O). **b**, TEM image and size distribution histogram of Au-TDASiW9. **c**, The results of catalytic aerobic oxidation of **1a**.

Therefore, photographs of the reaction solutions, TEM images and catalytic test results of POM-protected gold nanoparticles using THA, TDA and CTA as cations were added in Supplementary Fig. 20, and relevant contents were added to the revised manuscript as follows:

“To investigate the effect of cations, preparation of gold nanoparticles protected with SiW9 was examined using different cations such as tetrahexylammonium (THA), tetradecylammonium (TDA), and cetyltrimethylammonium (CTA) instead of TOA. Although TDA can be used for preparation of gold nanoparticles, incomplete phase transfer was observed for THA and CTA owing to less hydrophobicity of metal precursors and emulsion formation, respectively (Supplementary Fig. 20a). The obtained gold nanoparticles using TDA cations and SiW9 (Au-TDASiW9) possessed similar particle sizes of 3 nm and similar catalytic reactivity to those of Au-TOASiW9, suggesting that countercations did not necessarily facilitate the reaction (Supplementary Fig. 20b, c). These results provide direct evidence that multi-dentate POM ligands do not only contribute to stabilising the small (~3 nm) gold nanoparticles during their preparation but also allow retaining their catalytically active sites, enabling the modulation of the electronic states of gold nanoparticles for activity control.”

(Page 12, Line 253)

Comment

Suzuki and co-workers from Au(0) nanoparticles in toluene by borohydride reduction of H₂AuCl₄ in the presence of [SiW₉O₃₄](10⁻), (SiW₉) and a large excess of tetra-n-alkylammonium bromide (TOAB). The material is then characterized mostly in after drying by a routine spectroscopic and microscopic methods and used under strongly basic conditions, i.e., after addition of alkali-metal cations salts of carbonate, CO₃(2⁻) as aerobic catalysts for the oxidation of benzyl alcohol to benzaldehyde, and for oxidations of related substrates. The authors also characterize the properties of the final product by comparison to synthetic results obtained starting from SiW₁₂O₄₀(4⁻), SiW₁₁O₃₉(8⁻) and SiW₁₀O₃₆(8⁻). The claimed findings are significant, but considerable work is needed to correct clear errors in understanding the stability of the inorganic clusters and in demonstrating the structures and compositions of the clusters present after treatment with borohydride in synthesis and carbonate in catalysis. Moreover, some key experimental data critical to interpreting the results are missing from the Methods section of the text as well as from the supplementary information. Without that information (specified below), some of the reported results cannot be properly evaluation. Regarding the catalytic studies, additional control experiments, as well as kinetic and analytical data are needed to provide sufficient evidence for claimed activity and stability. Overall, the above items need to be fully addressed in order to qualify the work for publication in any quality journal of chemistry, inorganic chemistry or catalysis.

Response

Thank you very much for providing many valuable comments. We have carefully considered all your comments and revised the manuscript. Since many of the comments in the previous submission were very meaningful, the quality of our paper has now been improved significantly. Please confirm the following point-to-point responses.

Comment

An additional problematic issue relates to scholarship and context, i.e., representation of the work in the context of closely related previously-published studies; those very relevant studies are entirely neglected. For example, the closely related work of Richard Finke who prepared organic-solvent soluble catalysts very similar to those reported and proposed here are highly stable and much more reactive than those reported here. However, that entire opus of work, including complexes of Rh and Ir is ignored by the present authors. This is even more problematic in that the present authors propose using POMs in organic solvent to prepare stable colloids of "platinum, ruthenium, rhenium and rhodium", again with no mention of the fact that Finke prepared highly stable and reactive complexes of Rh and Ir over 25 years ago, and published numerous detailed high-quality articles on the "electrosteric stabilization" he observed. For example, the authors refer to the fact that POMs "have emerged as efficient protecting agents for stabilising metal nanoparticles by means of efficient coordination, electrostatic repulsion and steric hindrance". The latter was discovered by Finke but not a single article from his 15 years of work on this topic is cited by the authors.

Response

We totally agree that the field of POM-modified metal particles has been developed by pioneering researchers, such as Finke, Nadjo, Weinstock, Neumann, Kulesza, Wang and so on. In our recent review paper (*Angew. Chem. Int. Ed.* **2023**, *62*, e202214506), we emphasized that the research field of metal nanoparticle/POM hybrid materials emerged after the demonstration that POMs could modify electrode surfaces (mercury, platinum and gold) by Nadjo in 1980s (*J. Electroanal. Chem. Interfacial Electrochem.* **1985**, *191*, 441–448), and very early-stage work of using photoreduced POMs to reduce Ag⁺ by Chalkley in 1952 (*J. Phys. Chem.* **1952**, *56*, 1084–1086), and the first preparation of POM-modified metal nanoparticles (Ir and Rh) by Finke through H₂ reduction of iridium and rhodium complex in 1994. In the pioneering works of Finke and co-workers, many solid findings were observed including the existence of both reactivity and durability of iridium and rhodium nanoparticles in hydrogenation reactions (*J. Am. Chem. Soc.* **1994**, *116*, 8335–8353; *Inorg. Chem.* **1994**, *33*, 4891–4910; *J. Am. Chem. Soc.* **1997**, *119*, 10382–10400; *J. Am. Chem. Soc.* **1998**, *120*, 9545–9554).

It should be pointed out that the use of 1,5-cyclooctadiene was necessarily employed for metal precursors (e.g., iridium and rhodium) in Finke's method, while such strategy was not reported for the gold nanoparticles. Owing to possible weaker interactions between POMs with gold than other metals, there are few reports of preparation of gold nanoparticles in organic solvents except a recent report by Cronin using a microwave-assisted synthesis in a polar organic solvent, acetonitrile (*Inorg. Chem.* **2019**, *58*, 4110–4116). However, much larger gold nanoparticles (20–100 nm in diameters) formed using various types of POM ligands in acetonitrile in comparison to gold nanoparticles (<20 nm) prepared in aqueous media. In our preliminary findings, the precipitation of gold nanoparticles was immediately observed in organic solvents such as acetone, acetonitrile, pyridine, methanol, by addition of tetrabutylammonium borohydride to the mixing solution of chloroauric acid and tetrabutylammonium salts of **SiW9** ((C₁₆H₃₆N)₄H₆SiW₉O₃₄), **TBASiW9**). These results and findings strongly indicate that to develop ultrastable and reactive gold nanoparticles is different from and likely more difficult than previously reported iridium and rhodium nanoparticles.

Furthermore, our standing point of using non-polar organic solvent (not merely organic solvent) is that multidentate POMs are known to interact with polar solvent molecules (e.g., hydration in water and coordination in methanol, etc.), which eventually leads to destabilization of metal nanoparticles during storage and/or usage in these polar solvents. Then, since isolation and crystallization of soluble salts in toluene are difficult, we employed a phase transfer method in this work by utilizing the cation exchange process of POMs (and metal precursors), followed by simple reduction with sodium borohydride to obtain POM-modified metal nanoparticles in toluene. Based on the above efforts, ultrastable and highly reactive gold nanoparticles can be obtained and directly applied in diverse catalytic reactions for the first time.

Based on your advice, regarding "have emerged as efficient protecting agents for stabilising metal nanoparticles by means of efficient coordination, electrostatic repulsion and steric hindrance", the relevant citations and contents about Finke's pioneering works were added as follows:

"have emerged as efficient protecting agents for stabilising metal nanoparticles by means of efficient coordination, electrostatic repulsion and steric hindrance (sometimes called as electrosteric stabilization, Fig. 1a and Supplementary Table 1, Entries 13–24).^[27–37]"

(Page 2, Line 44)

Added references:

[27] Lin, Y. & Finke, R. G. Novel polyoxoanion- and Bu₄N⁺-stabilized, isolable, and redissolvable, 20–30-Å Ir_{300–900} nanoclusters: the kinetically controlled synthesis, characterization, and mechanism of formation of organic solvent-soluble, reproducible size, and reproducible catalytic activity metal nanoclusters. *J. Am. Chem. Soc.* **1994**, *116*, 8665–8353.

[28] Aiken, J. D. & Finke, R. G. Nanocluster formation synthetic, kinetic, and mechanistic studies. The detection of, and then methods to avoid, hydrogen mass-transfer limitations in the synthesis of polyoxoanion- and tetrabutylammonium-stabilized, near-monodisperse 40 ± 6 Å Rh(0) nanoclusters. *J. Am. Chem. Soc.* **1998**, *120*, 9545–9554.

Comment

Similarly, Ira Weinstock published numerous articles on stabilization of gold nanoparticles using mono-defect Keggin POM anions. Like Finke's POM-stabilized NPs prepared in organic solvents, Weinstock's are stable for years without aggregation in water and include examples of very small particles, 2-3 nm. Yet, in Figure 1a, the authors summarize "polyoxometalate protection" by showing stabilization of Au NPs by plenary Keggin anions (as opposed to the mono-defect ones used in the works noted above) and then claim that they "agglomerate during storage" and are limited to "relatively large particle sizes". This reference to only plenary POMs, which are quite rare, if reported at all, as the context for their current findings is disturbingly misleading.

Response

We totally agree that Weinstock has published many important works regarding POM-protected gold nanoparticles. Especially, his group reported utilization of cryo-TEM to observe POM monolayers on

gold nanoparticles and developed a series of effective methods in studying surface states and structures of POM-protected metal (oxide/hydroxide) nanoparticles in aqueous solution (*J. Am. Chem. Soc.* **2009**, *131*, 17412–17422; *ACS Nano* **2012**, *6*, 629–640; *Nat. Commun.* **2018**, *9*, 4896; *ACS Catal.* **2021**, *11*, 11385–11395). In addition, he reported the preparation of small gold nanoparticles of several nanometers with POM ligands (*Angew. Chem. Int. Ed.* **2017**, *56*, 7083–7087), and these small gold nanoparticles can be subsequently used to obtain gold supraspheres (diameter ca. 150 nm) that can be stable in water for several weeks towards application in host-guest chemistry (*Nat. Nanotechnol.* **2017**, *12*, 170–176).

Whereas, referring to the reports on gold nanoparticles by Weinstock, stabilities of small gold nanoparticles of diameters within several nanometers have not been specifically mentioned, to the best of our knowledge. Alternatively, there are three reports discussing the stabilities of larger gold nanoparticles: i) 14 nm gold nanoparticles with $K_9AlW_{11}O_{39}$ obtained by ligand exchange from citrate can be stable after one month at room temperature (*J. Am. Chem. Soc.* **2009**, *131*, 17412–17422); ii) 13.8 nm gold nanoparticles co-protected by alkanethiolate and $[AlW_{11}O_{39}]^{10-}$ can be relatively stable after heating at 80 °C for 4 h (*ACS Nano* **2012**, *6*, 629–640); iii) 150 nm gold nanoparticles protected by $[AlW_{11}O_{39}]^{9-}$ supraspheres can be stable in water for several weeks and 200 nm Au- $[AlW_{11}O_{39}]^{9-}$ supraspheres precipitate from solution within 24 h, and longer storing period for at least six month can be achieved by replacing POMs with thiolates (*Nat. Nanotechnol.* **2017**, *12*, 170–176). Together with necessary immobilization procedures of colloidal metal rather than just gold nanoparticles on supports such as alumina, carbon, graphene oxide, cadmium sulfide in previous reports (*Angew. Chem. Int. Ed.* **2023**, *62*, e202214506), we acknowledge that POM protection can enhance stability of colloidal metal nanoparticles. However, towards practical catalytic application under conditions such as high metal concentration, long time aging, heating, addition of base and organic substrate, there has still been lacked a feasible and universal methodology of solving “semi-stability issue” of POM-protected metal nanoparticles, which hinders further advance of these interesting materials.

In our preliminary experiments, even by using multidentate POM ligands such as **SiW9**, gold nanoparticles have also been found subject to agglomerate quickly under catalytic conditions, such as heating (50 °C), pH (for aqueous solution), and even precipitate immediately from aqueous solution, either upon addition of organic substrate such as 1-methyl-4-piperidone for the catalytic dehydrogenation of piperidone derivatives (*Angew. Chem. Int. Ed.* **2022**, *61*, e202205873), or under high metal concentration conditions (>2.5 mM). Based on these experimental findings and as-conducted literature review, we would like to underline that “semi-stability issue” is an inevitable problem in conventional aqueous-media-based synthetic methods regardless of the types of POMs. Thus, following our previous engagement on organic-solvent-based synthesis of POMs (*Coord. Chem. Rev.* **2022**, *469*, 214673), and in light of Cronin’s exploration on polar organic solvent (*Inorg. Chem.* **2019**, *58*, 4110–4116), we developed this non-polar-solvent-based multi-dentate POM protection strategy.

In order to avoid possible misunderstandings, advantages and disadvantages for POM protection for relevant contents in introduction and Fig. 1 were revised, and relevant citations about Weinstock and Wang’s pioneering works were added as follow:

Added references:

[29] Wang, Y., Neyman, A., Arkhangelsky, E., Gitis, V., Meshi, L. & Weinstock, I. A. Self-assembly and structure of directly imaged inorganic-anion monolayers on a gold nanoparticle. *J. Am. Chem. Soc.* **131**, 17412–17422 (2009).

[30] Wang, Y., Zeiri, O., Neyman, A., Stellacci, F. & Weinstock, I. A. Nucleation and island growth of alkanethiolate ligand domains on gold nanoparticles. *ACS Nano* **6**, 629–640 (2012).

“However, POM-protected gold nanoparticles sometimes undergo agglomeration in solution during storage and/or usage,^{31,32,36,37} which can be attributed to decomposition and structure transformation of POMs in the commonly used aqueous media or occupation of the vacant sites of POMs by alkali metal cations and solvent molecules, leading to destabilization of gold nanoparticles.³⁸⁻⁴¹ Moreover, the restriction to hydrophilic use besides the semi-stability issue of POM-protected gold nanoparticles has

also limited the exploration of such a feasible molecular-level catalyst design towards practical catalytic applications.³²”

(Page 2, Line 50)

Fig. 1 | Preparation of gold nanoparticles. a, Representative methods for the preparation of gold nanoparticles using thiol protection, organic polymer protection and polyoxometalate protection. b, This work: a non-polar-solvent-based multi-dentate polyoxometalate protection method for developing ultra-stable and highly reactive gold nanoparticle catalysts.

Comment

Finally, Ronny Neuman has published numerous articles on various different metal NPs stabilized in both water and organic solvent by PV2Mo10O40(5-), and has shown many of those to be highly stable and remarkably reactive and selective catalysts for aerobic oxidations under mild conditions.

Response

In our recent review paper (*Angew. Chem. Int. Ed.* **2023**, 62, e202214506), we have underlined that Neumann has done many pioneering works on catalysis of POM-protected metal nanoparticle. Regarding

exploration on colloidal metal nanoparticle catalysts by Neumann, POM-stabilised Pd nanoparticles were prepared in acetophenone and α,α,α -trifluorotoluene, and applied them to C–C and C–N coupling reactions and aerobic oxydehydrogenation (*Org. Lett.* **2002**, *4*, 3529–3532; *Adv. Synth. Catal.* **2007**, *349*, 1624–1628). Whereas, in the former one, $K_5PPdW_{11}O_{39}$ was required to be the precursors, and relatively large particles (15–20 nm) were obtained (*Org. Lett.* **2002**, *4*, 3529–3532); in the later one, pre-functionalisation of $[SiW_{11}O_{39}]^{9-}$ with alkanethiol tether was necessary, while in contrast, Pd precipitates were formed using just $[SiW_{11}O_{39}]^{9-}$, suggesting superior protection effect of thiolate rather than POMs (*Adv. Synth. Catal.* **2007**, *349*, 1624–1628). Despite unique and high reactivity of as-obtained Pd nanoparticles, these factors have limited further exploration of materials and their catalysis.

Later, by utilizing $[PV_2Mo_{10}O_{40}]^{5-}$ ligands, they reported the preparation of Ag, Ru, Rh, Ir, Pt nanoparticles in aqueous solution (*Chem. Commun.* **2005**, 4595–4597; *Catal. Lett.* **2008**, *123*, 41–45). However, those POM-protected metal nanoparticles ought to be immobilized on alumina supports before catalytic reactions at relatively harsh reaction conditions, such as direct aerobic epoxidation of alkenes (160 °C, 1 h, *Chem. Commun.* **2005**, 4595–4597) and aerobic oxidation of secondary alcohols (125 °C, 14 h, *Catal. Lett.* **2008**, *123*, 41–45). Furthermore, in spite of the immobilization on alumina supports, decomposition of POMs was also observed after the catalytic reactions (*Chem. Commun.* **2005**, 4595–4597).

Based on the above discussions, we conclude that regardless of significance of Neumann's pioneering studies, the preparation methodology limitation to specific metal/POM precursors and essential pre-functionalisation, necessary immobilization on solid supports, and lack of exploration on gold-based catalysis make it less relevant to research focus in this work.

Comment

In addition to all the above, Pavel Kulesza has used phase transfer methods to prepare POM stabilized metal NPs in organic solvents. Not a single citation of any article from Kulesza's opus of work on that topic is cited by the present authors.

Response

Before the development of a ligand exchange method from citrate-protected gold nanoparticles to POM-protected ones in aqueous solution by Weinstock and co-workers (*J. Am. Chem. Soc.* **2009**, *131*, 17412–17422), Kulesza and Cox reported the ligand exchange process from alkanethiol-protected gold nanoparticles in non-aqueous solution (e.g., hexane) to POM-protected gold nanoparticles in sulfuric acid solution (*Electroanalysis* **2007**, *19*, 2103–2109). However, despite obtaining small gold nanoparticles of diameters within several nanometers, the final POM-protected gold nanoparticles were still in aqueous solution suffering from as-mentioned “semi-stability issue”, limiting their catalysis. Then, highly acidic conditions were required to ensure “thermally-unfavorable” exchange from thiolates to POMs, thus restricting to very specific types of POM ligands. Next, in comparison to ligand exchange process that may be accompanied by complexity in purification, characterization and proper evaluation of residue ligands, direct preparation of POM-protected gold nanoparticles can avoid such concerns. Finally, the only requirement in this method is the “cation exchange to finish phase transfer”, that is a common feature for various metal and POM precursors, making it possible to execute systematic studies. Thus, relying on either materials properties or methodology procedures, this methodology by Kulesza and Cox can be considered as ligand exchange process taking place among two phases, that is less relevant to the synthetic methodology in this work but worth underlining as representative synthetic routes.

Based on above discussions, Supplementary Table 1 was revised and Kulesza's phase-transfer method and Weinstock's one-pot synthesis method were added as Entry 17 and 21, respectively.

Comment

The aerobic catalysis based on dioxygen activation by POM stabilized Au NPs reported here is also not new, as similar O₂ activation by POM-stabilized gold NPs was reported a few years ago by Wang and co-workers (ref. 29).

Response

It has been theoretically studied that O₂ can be efficiently activated on negatively charged (or called as anionic) gold surface (*J. Phys. Chem. A* **2018**, *122*, 3346–3352; *Catal. Sci. Technol.* **2021**, *11*, 3333–3346), and not only organic polymers such as poly(*N*-vinyl-2-pyrrolidone) (PVP) but also POM-protected gold nanoparticles have been confirmed efficacy in aerobic alcohol oxidation or carbon monoxide oxidation (*J. Am. Chem. Soc.* **2009**, *131*, 7086–7093; ref 29: *Inorg. Chem.* **2017**, *56*, 2400–2408). Particularly, in the work by Wang and Weinstock, a direct role of the POM ligands (electronic interaction) in modifying the catalytic activities (e.g., CO oxidation) of the metal(0) cores themselves was for the first time documented, while in comparison, either simply providing a stabilizing function, or the reversible redox properties of the POM ligands in quenching organic-radical intermediates were reported in other reports (*Inorg. Chem.* **2017**, *56*, 2400–2408). In the notable works of “anionic” gold nanoparticles in catalysis, effective activation of O₂ on electron-rich gold surface was found as the key in enhancing reactivity for CO oxidation.

However, owing to stability issue of the above-mentioned anionic gold nanoparticles under typical catalytic reaction conditions, such as high temperature, high concentration, and addition of organic substrates and bases, how anionic gold nanoparticles behave in practical catalytic organic functional group transformations has not been explored (or reported) until our recent work of POM-protected gold nanoparticles immobilized on carbon support (*Angew. Chem. Int. Ed.* **2022**, *61*, e202205873).

The present work provides a further great advance in this research area of developing ultrastable and highly reactive gold nanoparticles, and they can be capable of directly used in solution phase to not only well-perform in the typical model reaction of aerobic alcohol oxidation like Au-PVP, but also distinct catalytic properties in dehydrogenation of piperidone derivations and cross-dehydrogenative-coupling reactions. These unique catalytic properties of POM-protected metal (e.g., gold) nanoparticles are interesting and yet not reported before. We believe that they are also merely found in previous researches of analogously heterogeneous catalysis and should hold great potentials in development of novel organic functional group transformations considering wide applicability of this methodology shown in this work.

Comment

In the context of the above works, published by several leading groups over many years, the conceptually new finding reported here is effectively disclosure of a synthetic method, demonstrated for a single type of metal nanoparticle (gold). This would be abundantly apparent if through proper scholarly presentation that context were properly included in Introduction and Discussion of the present article. In the context of the above-discussed studies, the new synthetic method reported here and used for a single example (Au) does not justify publication in Nature Commun.

Response

Based on the above explanations, despite numerous reports in search of co-existence of both reactivity and stability of metal nanoparticles with POM ligands (Fig. R1a, c, e), a general methodology that can retain high stability and reactivity during catalytic applications is still in demand (Fig. R1a-f). Then, possibly owing to relatively weaker interaction between POM ligands and gold (and silver) nanoparticles, there has not been reported the synthesis of highly reactive gold nanoparticles that even maintain stable in different harsh reaction conditions (Fig. R1a, c). Furthermore, although POM protection has been confirmed with outstanding stabilization effect and electronic effect essential for catalysis, their potentials in catalyst design for organic functional group transformations remains to be explored (Fig. R1a-f).

In this work, based on our previous knowledges that multidentate POMs ought to strongly interact with metal (e.g., gold) nanoparticles, and key concerns that interruption of counter cations and/or solvent molecules needs to be avoided, a non-polar-solvent-based strategy was developed in obtaining ultrastable and highly reactive gold nanoparticles for the first time (Fig. R1g). As-obtained gold nanoparticle catalysts were further found to possess unique catalytic behaviors and more importantly, there exist no specific restrictions to metal/POM precursors and preparative procedures expanding to various metals, POMs, cationic surfactants, and organic solvents, qualifying it a necessary and important part of investigating and utilizing metal nanoparticle-POM hybrid materials in diverse research fields including catalysis,

energy conversion, coordination chemistry, biochemistry, and materials science. Hence, we believe that the unusual but essential co-existence of reactivity, durability and applicability of colloidal metal nanoparticles developed in this work justify the potential publications in Nature Communication both from advanced scientific significance and wide readership community.

a) Finke’s work: Ir and Rh nanoparticles protected with P2W15Nb3 (1990s)

- Organometallic precursors (COD) required H₂ reduction-capable metals

b) Papaconstantinou’s work: Au, Ag, Pt, Pd nanoparticles protected with SiW12 (2002)

- Sacrificing agent required Photoreactive polyoxometalates with low redox potential

c) Neumann’s work: Ag, Ru, Rh, Ir, Pt nanoparticles protected with PV2Mo10 (2000s)

- Redox reaction of Zn and H₅PV₂Mo₁₀O₄₀ required Essential immobilization on α-Al₂O₃

d) Kulesza’s work: Au nanoparticles protected with PW12 by ligand exchange (2007)

- Complex preparation procedures Essential immobilization on supports

e) Weinstock’s work: Au nanoparticles protected with KAIW11 (2009 & 2017)

i) Ligand exchange route

ii) Direct reduction route

- Catalytic limitation to oxidation of carbon monoxide (room temperature)

f) Cronin’s work: Au nanoparticles protected with W12 (2019)

- Specific metal precursors required Fast agglomeration/aggregation issue

a-f) remaining issues based on previous reports till now (1994–2023)

- Universal protocol with few restrictions to metal and ligand precursors to be developed
 Most reports limiting to aqueous synthesis and hydrophilic use
 Stability to be improved under conditions such as concentration and heating

g) This work: Au, Pt, Ru, Rh, Re nanoparticles protected with SiW9 (2023)

- ✓ Stability in high concentration, heating, aging, and addition of base
 ✓ Applicability to various components (no specific restrictions)
 ✓ Reactivity in various catalytic organic reactions ✓ Hydrophobic use possible

Fig. R1 | Comparison of this method to representative methods on preparation of polyoxometalate-protected metal nanoparticles.

Comment

In addition to the above-noted issues of scholarly presentation, there are many points of science that need to be addressed before this work is perhaps submitted elsewhere. These are enumerated below.

I. The stability of SiW9 on Au after synthesis and after catalysis

During the synthesis, the authors combined HAuCl₄ with SiW₉ and then added NaBH₄. Both operations can decompose the POM. It is also well-known that SiW₉ will break down in the presence excess Na₂CO₃. Therefore, it is likely that a large amount of SiW₉ decomposed during the catalytic applications.

To address the above issues, the authors should give adequate proof for the stability and presence of pure SiW₉ both after synthesis and after catalysis. Notably, no analytical data whatsoever in support of POM stability after catalysis is provided. The authors only measure particle size by TEM! The following experiments are clearly needed.

Response

It has been known that undesired decomposition or transformation of lacunary POMs sometimes occur in aqueous media (*Coord. Chem. Rev.* **2022**, *469*, 214673). In our recent efforts of addressing this stability issue, we employed alkylammonium salts of lacunary POMs in organic media to prepare stable and reactive metal(-oxo) nanoclusters even in practical conditions such as heating, addition of strong acid/base and catalytic use (*Chem. Sci.* **2022**, *13*, 5557–5561; *Chem. Eur. J.* **2022**, *28*, e202104051; *Nat. Chem.* **2023**, *15*, 940–944). In addition, we recently reported interactions of polar organic solvent molecules and lacunary POM ligands (*J. Am. Chem. Soc.* **2019**, *141*, 7687–7692; *Inorg. Chem.* **2022**, *61*, 9841–9848). Therefore, in this work, to suppress the structure transformation of lacunary POMs and interaction with solvent molecules, we investigated the development of POM-protected metal nanoparticles in non-polar solvents by sequential phase transfer process of metal precursors (e.g., chloroauric acid) and alkali metal salts of POMs into toluene phase, followed by reduction of metal precursors using sodium borohydride.

In this work, we employed IR, Raman and XAFS (XANES and EXAFS) analysis to investigate the structures of lacunary POMs of POM-protected gold nanoparticles in detail.

From IR spectra, the characteristic peaks of Au-TOASiW₉ from 800 to 1000 cm⁻¹ (i.e., adsorption peaks located around 810, 880, 940 and 980 cm⁻¹) were similar to those of TOASiW₉ (Supplementary Fig. 11), supporting the preservation of the structure of SiW₉. Although the IR spectrum of TOASiW₉ was slightly different from that of NaSiW₉ (sodium salt of SiW₉), similar phenomena were also observed in previous studies by Newton in their K₆[P₂W₁₈O₆₂]-protected gold nanoparticles and our Na₁₀[SiW₉O₃₄]-protected gold nanoparticles, possibly originating from “dynamic assemblies” nature of POM protection (*Angew. Chem. Int. Ed.* **2020**, *59*, 14331–14335; *Angew. Chem. Int. Ed.* **2022**, *61*, e202205873).

According to Raman spectra, apart from peaks being assigned to toluene and TOAB at 760, 775, 785, 890, 1005, 1115, 1130, 1140 and 1170 cm⁻¹, the characteristic peak of W=O_d bonding from POM structures was observed at 965 cm⁻¹ for TOASiW₉ and Au-TOASiW₉ even after alcohol oxidation, similar to that of the tetrabutylammonium salt of SiW₉ (TBA₄H₆SiW₉O₃₄, TBASiW₉) at 970 cm⁻¹ but slightly shifted from that of the sodium salt of SiW₉ (Na₁₀SiW₉O₃₄, NaSiW₉) at 940 cm⁻¹ (Supplementary Fig. 12). This phenomenon can be ascribed to the presence of sodium cations near the POM anions in the case of NaSiW₉ increasing the W=O bonding length and weakening the bonding strength (*J. Phys. Chem. B* **2000**, *104*, 8160–8169; *Nanoscale* **2012**, *4*, 502–510). These peaks were totally different from those of sodium tungstate (Na₂WO₄), also supporting that decomposition of SiW₉ did not occur (Supplementary Fig. 12).

Furthermore, the XAFS study was performed. As it was presented in the manuscript that similar patterns of Au-TOASiW₉ and TOASiW₉ in the second derivate of white-line region and W L₃-edge *k*-space EXAFS spectra, indicating that POMs maintained their structures after hybridization with gold nanoparticles (Figure 3e, f). Then, similar patterns of TOASiW₉ and NaSiW₉ but completely different from those of KSiW₁₂ and WO₃ in the second derivate of white-line region indicated that there existed no obvious structural changes in the {WO₆} octahedra (Supplementary Fig. 13). The W L₃-edge *k*-space EXAFS spectra showed no significant changes between TOASiW₉ and NaSiW₉ while in contrast to those of the potassium salt of SiW₁₂ (K₄SiW₁₂O₄₀, KSiW₁₂) and WO₃, strongly indicating that POMs maintained intact structures as well. In the *R*-space EXAFS spectra, the peaks at *R* = 1.2 and 1.7 Å and 3.2 Å assignable to terminal W=O and bridging W–O–W and W–W, respectively, exhibited no drastic

changes from **NaSiW9** to **TOASiW9**, further supporting the preservation of POM structures.

In addition to the above characterization results confirming POM structures in toluene solution, in this revision, we also investigated the structures of POMs that were deliberately transferred into aqueous phase by using cation exchange property of POMs. In the IR spectra, the characteristic peaks of **SiW9** in the region of 500–1000 cm^{-1} were well consistent between **NaSiW9** and POMs after mixing with gold precursors and sodium borohydride, supporting the intact **SiW9** structure in this method.

Based on the above discussions and findings, we added Raman spectrum of Au-**TOASiW9** after the catalytic reaction in Supplementary Fig. 12, and comparison of XAFS studies of **TOASiW9** with other materials in Supplementary Fig. 13, and Supplementary Fig. 14 for confirming POM structures during synthesis. Accordingly, we revised and appended the manuscript as follow:

“Infrared (IR) spectroscopies showed that Au-TOASiW9 exhibited similar bands to those of TOASiW9 and NaSiW9 regarding characteristic peaks in the region from 800 to 1000 cm^{-1} , but differed from those of sodium tungstate, indicating that the structure of SiW9 was preserved (Supplementary Fig. 11).^{34,36} According to Raman spectroscopies, the characteristic peak of $W=O_d$ bonding from POM structures was observed at 965 cm^{-1} for Au-TOASiW9 and TOASiW9, similar to that of the tetrabutylammonium salt of SiW9 (TBA₄H₆SiW₉O₃₄, TBASiW9) at 970 cm^{-1} but slightly shifted from that of the sodium salt of SiW9 (Na₁₀SiW₉O₃₄, NaSiW9) at 940 cm^{-1} (Supplementary Fig. 12). This can be ascribed to the presence of sodium cations near the POM anions in the case of NaSiW9 increasing the $W=O$ bonding length and weakening the bonding strength.⁴⁵”

(Page 8, Line 161)

“Similar patterns of Au-TOASiW9 and TOASiW9 in the second derivate of white-line region and $W L_3$ -edge k -space EXAFS spectra, indicating that POM maintain their structures after hybridization with gold nanoparticles (Fig. 3e, f).^{34,44} Then, similar patterns of TOASiW9 and NaSiW9 but completely different from those of the potassium salt of SiW12 (K₄SiW₁₂O₄₀, KSiW12) and WO₃ in the second derivate of white-line region indicated that there existed no obvious structural changes in the $\{WO_6\}$ octahedra (Supplementary Fig. 13a, b). The $W L_3$ -edge k -space EXAFS spectra showed no significant changes between TOASiW9 and NaSiW9 while in contrast to that of KSiW12 and WO₃, strongly indicating that POM maintained intact structures as well (Supplementary Fig. 13c). In the R -space EXAFS spectra, the peaks at $R = 1.2, 1.7 \text{ \AA}$ and 3.2 \AA assignable to terminal $W=O$, bridging $W-O-W$ and $W-W$, respectively, exhibited no drastic changes from NaSiW9 to TOASiW9, further supporting the preservation of POM structures in this method (Supplementary Fig. 13d). Finally, structures of POMs during the synthesis of gold nanoparticles were confirmed through deliberately transferred into aqueous phase (Supplementary Fig. 14, Supplementary Table 2, Entry 5). In the IR spectra, the characteristic peaks of SiW9 in the region of 500 – 1000 cm^{-1} were well consistent between NaSiW9 and POMs after mixing with gold precursors and sodium borohydride respectively, indicating their intact structures in current method (Supplementary Fig. 14). These results demonstrate that POM ligands remain stable in this synthetic system, effectively protecting the metal nanoparticles.”

(Page 9, Line 170)

Supplementary Fig. 11 | IR spectra of Au-TOASiW9, TOASiW9, NaSiW9, TOAB and KSiW12.

Supplementary Fig. 12 | Raman spectra of Au-TOASiW9, Au-TOASiW9 after the catalytic oxidation of 1a, TOASiW9, NaSiW9, TBASiW9, TOAB and sodium tungstate (Na_2WO_4).

Supplementary Fig. 13 | XAFS studies of TOASiW9, NaSiW9, WO₃ and KSiW12: **a**, W-L₃-edge XANES spectra in wide range. **b**, Second derivatives in white line area. **c**, k^3 -Weighted W L₃-edge EXAFS spectra. **d**, Fourier-transformed R-space EXAFS spectra (overlapping of red and black line at $R = 1.2$ Å).

a Illustrative scheme of deliberately transferring POMs back to aqueous phase

b Sample preparation and FTIR characterization of POM samples

Supplementary Fig. 14 | Confirmation of structures of POMs in aqueous phase during synthesis after mixing with gold precursors and addition of sodium borohydride. **a**, Illustrative scheme of operative procedure. **b**, Photographs of solid samples by freeze-drying aqueous phase containing POMs and their IR spectra in comparison to **NaSiW9**.

Comment

1. Report the ESI-MS spectrum of the colloidal solution to verify if other POMs are formed. The IR in Figure S10 actually indicates that a large fraction of SiW_9 decomposed.

Response

We agree that ESI-MS is one of the important characterization methods of POMs. However, our efforts to obtain ESI-MS spectra of POMs in various conditions were failed, and we only observed peaks of TOA and bromide as $([\text{TOA}]_x[\text{Br}]_{x+1})^-$ (Fig. R2). The presence of TOAB made it difficult to observe POMs by ESI-MS. Alternatively, we carried out IR, Raman, W L_3 -edge XANES and EXAFS for the characterization of POMs as discussed above.

Fig. R2 | ESI-MS spectra for Au-TOASiW9 in mixing solution of toluene and acetonitrile (toluene/acetonitrile = 1/1, v/v).

Comment

2. The authors should purify the colloidal solution using either by dialysis to remove excess POM, or by ultracentrifugation-redispersion to obtain a pure POM-stabilized AuNPs. Check the activity and stability of the purified sample.

Response

According to your suggestion, centrifugation-redispersion and dialysis were examined. However, by centrifugation at 20000 rpm, no precipitation was observed, and it was difficult to isolate pure POM-stabilized gold nanoparticles. Then, a semi-permeable membrane made of regenerated cellulose that can be resistant to toluene was bought, and employed in dialysis, however, toluene solution was found incapable to be separated owing to compatibility issues.

We think that the coordination of POMs on the surface of gold nanoparticles is reversible and in equilibrium, and the stabilization of gold nanoparticles require excess POM ligands. Therefore, these purification procedures are technically difficult and of less practical necessity here.

Comment

3. The authors can readily provide evidence for that POM ligands are bound to gold as shown in their graphics. For this, they simply need to titrate their solutions by addition of thiol determine whether they reach a definitive endpoint. That will also provide quantitative data concerning how many POMs are bound to each gold particle, on average.

Response

According to your comments and previous findings of elucidating ligand exchange process from POM to thiol ligands through changes of surface plasmon resonance (SPR) by Weinstock (*ACS Nano* **2012**, *6*, 629–640), we carried out analogous experiments by dropwise adding dodecanethiol to the toluene solution of Au-TOASiW9. The UV-vis spectra of titration exhibit the continuous decrease of SPR band with showing isosbestic points upon addition of toluene solution of dodecanethiol, and then become constant after addition of 70 μL of the solution (dodecanethiol concentration, 1.56×10^{-4} mol/L). These results indicate that the ligand exchange from POMs to dodecanethiols occurs.

Based on the density and molar mass of Au as 19.3 g/cm^3 and 197 g/mol respectively, assuming a spherical shape and a uniform face-centered cubic structure, for 3 nm gold nanoparticle, the average number of gold atoms is calculated as $N = \frac{\rho V}{M} \times N_A = \frac{\pi(3 \times 10^{-7})^3}{6} \times \frac{19.3}{197} \times 6.02 \times 10^{23} = 834$. Based on total concentration of gold atoms (from chloroauric acid as $5 \times 10^{-4} \text{ M}$, gold nanoparticle concentration is $6 \times$

10^{-7} M. Thus, for one gold particle, there were $\frac{1.56 \times 10^{-4}}{6 \times 10^{-7}} = 260$ dodecanethiol ligands. Then, based on previous knowledges that an average ratio as 8.5 during the ligand exchange process from mono-lacunary POMs ($[AlW_{11}O_{39}]^{9-}$) to 11-mercaptoundecanoate (*J. Am. Chem. Soc.* **2009**, *131*, 17412–17422; *ACS Nano* **2012**, *6*, 629–640), there should be around 30 POM ligands surrounding a 3 nm gold particle in this case, and surface coverage ratio can be also estimated as 47%, which can be reflected by using current illustrations of Au-TOASiW9 in Figs.1–4.

Based on the above discussions and findings, we added Supplementary Fig. 9 about titration experiments in confirming surface coverage of Au-TOASiW9, accordingly, we revised and appended the manuscript as follow:

“Through titration experiments of dodecanethiol to Au-TOASiW9 with inspired by previous reports,^{29,30} it was estimated that around 30 SiW9 ligands surrounded a 3 nm gold particle, and surface coverage can be estimated as 47% (Supplementary Fig. 9, see explanation in detail).”

(Page 9, Line 144)

Supplementary Fig. 9 | Titration experiments using dodecanethiol in confirming surface coverage of Au-TOASiW9: **a,b**, UV-vis spectra of Au-TOASiW9 (0.5 mM) in toluene upon addition of toluene solution of dodecanethiol (5 mM). **c**, A plot showing the change of the absorbance of SPR band (524 nm) upon addition of a toluene solution of dodecanethiol. **d**, Illustration for Au-TOASiW9.

Comment

4. The data in Supplementary Figure 5 suffer from the omission of key details needed to evaluate the finding. In panel a, the title catalyst, there is a large excess of TOA (10 fold) and of the POM in solution. This is compared with panel c, where small changes are observed for Au stabilized by TOAB. However, no where in the text or SI is it indicated the relative amount of TOA present in that sample. The immediate

question a reader asks here is why is that data omitted, and if the amount of TOA is less in the sample in panel c, perhaps the slightly lesser stability is attributed to that. The main problem here, again, is one of including all relevant information, with respect to scholarship as noted above and with respect to presentation of all relevant experimental details, as relevant to interpretation of the data in Figure S5.

Response

Regarding Supplementary Fig. 5, as you mentioned in panel c for Au-TOAB, the amount of TOA was also 10 fold which was already described in the Method section in detail. In addition, goldish colored solid on the wall of test tube was observed in panel c already for Au-TOAB which should be ascribed to the precipitation of gold nanoparticles.

Comment

5. In related work, the authors make a case of the role of the charge of SiW9 by preparing samples using SiW12O40(4-), SiW11O39(8-) and SiW10O36(8-). No data is provided to support the argument that these are intact after synthesis. Sure, the plenary ion is not; it readily converts to the lacunary when not in acidic conditions. The data in Figure 3d are indicated that something is not as assumed by the authors. Namely, The trend in binding energy with charge is not as the large blue arrow indicates. Rather, the binding energy reverses the trend at SiW10. The authors don't comment on this, but it is entirely consistent with some other POM or mixture of POMs present, either for that case or for the SiW12 and SiW11 cases that precede it in the figure. In summary, all new materials prepared and discussed must be fully characterized. In all cases, ESI-MS spectra of the claimed POMs must be provided, along with IR/Raman etc.

Response

In our previous work, the preparation of SiW9-protected gold nanoparticles was performed in aqueous solution (*Angew. Chem. Int. Ed.* **2022**, *61*, e202205873). We performed the reaction at 4 °C to suppress the structure change of lacunary POMs during the preparation of gold nanoparticles in aqueous solution, and the intact structures of lacunary POMs have been shown by XAFS, IR and Raman analysis in the previous study.

In the present study, we performed the preparation of POM-protected gold nanoparticles in toluene phase, and thus we can further suppress decomposition or structural change of lacunary POMs during the synthesis. The intact structure of SiW9 after the reaction was supported by detailed analysis using XAFS (XANES and EXAFS) as described above. To discuss the structures of other lacunary POMs, Raman characterizations were further carried (Fig. R3). The characteristic peaks of W=O_d bonding of Au-TOASiW10, Au-TOASiW11 and Au-TOASiW12 are similar to those of TOASiW10, TOASiW11 and TOASiW12, respectively, showing the intact structures of these lacunary POMs after the synthesis of gold nanoparticles. As for ESI-MS spectra, we could not be obtained spectra due to the presence of TOABr in the solution (see Fig. R2).

Regarding the XPS results (Fig. 3d), the trend in binding energy was Au bulk (84.0 eV) > Au-TOAB (83.2 eV) ≈ Au-TOASiW12 (83.1 eV) > Au-TOASiW11 (82.7 eV) ≈ Au-TOASiW10 (82.7 eV) > Au-TOASiW9 (82.3 eV), and no reverse of the trend at Au-TOASiW10 was observed. Similar trend was observed in catalytic activity in line with electronic states of gold nanoparticles, highlighting POM modification can be a feasible tool in controlling catalysis (Table 1). These significant difference in properties of hybrid nanoparticles also indicated that the structures of POMs were maintained in this method.

Fig. R3 | Raman spectra of silicotungstate-protected gold nanoparticles. **a.** Au-TOASiW10. **b.** Au-TOASiW11. **c.** Au-TOASiW12.

Comment

In catalysis, the authors report yields but don't comment on whether they checked for products that don't appear in the GC method.

6. As such, the authors should report H-1 NMR spectra of all product mixtures to verify claimed yields in catalysis.

Response

In the catalytic test, we adopted GC and GC-MS methods under conditions where all possible products such as benzaldehyde, benzoic acid, and ester can be observed, and used biphenyl as an internal standard for quantitative analysis. For example, in the Au-TOASiW9-catalysed aerobic oxidation of benzyl alcohol (**1a**), the possible products, such as benzaldehyde (**2a**), benzoic acid (**3a**), octyl benzoate (**4a**) and benzyl benzoate (**5a**) can be confirmed by the GC and GC-MS. By the detailed analysis, besides target product **2a**, formation of trace amount of octyl benzoate (**4a**) was confirmed (possibly formed by the reaction of **3a** and octylhalide), and the total carbon balance was over 95%, indicating that GC method adopted here was effective (Supplementary Table 3). In contrast, determination of product yields by ¹H NMR is difficult because of the presence of excess TOABr in the reaction solutions. Herein, the MS (EI) spectra of each aldehyde/ketone products were provided as Supplementary Fig. 21.

Based on the above discussions and findings, we added Supplementary Table 3 about full table of reaction products, and Supplementary Fig. 21 about MS (EI) spectra of aldehyde/ketone products in the revised SI.

Supplementary Table 3 | Selective aerobic oxidation of benzyl alcohol (1a) to benzaldehyde (2a) using different catalysts.^a

Entry	Catalyst	Conversion of 1a (%)	Yield (%)			
			2a	3a	4a ^b	5a
1	Au-TOASiW9	84	75	n.d.	4	n.d.
2	Au-TOASiW9 ^c	>99	92	n.d.	5	n.d.
3	Au-TOASiW10	23	17	n.d.	2	n.d.
4	Au-TOASiW11	26	24	n.d.	2	n.d.
5	Au-TOASiW12	6	4	n.d.	<1	n.d.
6	Au-TOAB	5	3	n.d.	<1	n.d.
7	Au-dodecanethiol	1	<1	n.d.	<1	n.d.
8	Au-dodecanethiol ^c	1	<1	n.d.	<1	n.d.
9	TOASiW9	<1	<1	n.d.	<1	n.d.
10	Au-TOASiW9 (Ar 1 atm) ^d	<1	<1	n.d.	<1	n.d.

^a Reaction conditions: **1a** (0.25 mmol), 3 mL toluene solution of colloidal gold nanoparticles (Au: 4 mol%), K₂CO₃ (0.5 mmol), room temperature (~25 °C), O₂ (1 atm), 24 h. ^b Formed by reaction of **3a** with octylhalide, decomposition product of TOAB. ^c Cs₂CO₃ (0.5 mmol) was used instead of K₂CO₃ (0.5 mmol). ^d Freeze-pump-thaw cycles were carried out and the reactor was connected to a balloon filled with an Ar gas.

Supplementary Fig. 21 | MS (EI) spectra of aldehyde/ketone products.

Supplementary Fig. 21 (continue)

Supplementary Fig. 21 (continue)

Comment

7. TOASiW9W11 should be added to sample reported in entry 6 (Table 1) to confirm the role of the POM in catalysis. The addition can be followed by UV-vis spectroscopy.

Response

Based on your comments, ligand exchange from surfactant-protected gold nanoparticles (Au-TOAB) to SiW9 was examined (Fig. R4). The change of SPR absorbance of gold nanoparticles upon addition of SiW9 indicated the ligand exchange. The obtained ligand-exchanged gold nanoparticles (Au-TOAB-SiW9) were applied to catalytic oxidation of benzyl alcohol, and significantly enhanced reactivity was found comparing to Au-TOAB but still lower than Au-TOASiW9 (Table R1, Entries 1–3). This can be contributed to the agglomeration of gold nanoparticles during ligand exchange as indicated by the increased absorbance around 700 nm of agglomerated gold nanoparticles (*ACS Nano* **2011**, *5*, 8070–8079; *Anal. Chem.* **2022**, *94*, 5310–5316).

In a similar manner, physical mixture of Au-dodecanethiol and TOASiW9 was examined with catalysis, in which the reaction merely proceeded similar to Au-dodecanethiol, probably owing to strong binding of dodecanethiol to the surface of gold nanoparticles (Table R1, Entries 4 and 5). Finally, this catalytic reaction did not proceed either at oxygen atmosphere by using TOASiW9 or at argon atmosphere by using Au-TOASiW9, indicating that oxygen atoms of POM ligands did not participate in the catalysis as well (Table R1, Entries 6 and 7).

It is also noted that no changes of SPR wavelength happened during ligand exchange process unlike either red-shift or blue-shift observed in aqueous solution (*ACS Nano* **2012**, *6*, 629–640, Supplementary Fig. 8, Fig. R4). Additionally, the presence of SPR after ligand exchange from Au-TOASiW9 to dodecanethiol, but the absence of SPR in Au-dodecanethiol were confirmed respectively (Supplementary Figs. 2, 5b, 8). Thus, based on i) agglomeration issue of gold nanoparticles, ii) possibly different nature between directly obtained and ligand exchanged samples, we think it is difficult to properly evaluate the effect of ligand exchange on catalytic reactivity and therefore use it for determining surface structures with certainty.

Based on above discussions and findings, we added catalytic test results of Au-TOASiW9 (Ar 1 atm) into Table 1 as Entry 10, accordingly, we revised and appended the manuscript as follow:

“Additionally, **TOASiW9** exhibited no activity and **Au-TOASiW9** did not promote the reaction under argon (Ar) atmosphere, confirming gold nanoparticles as the active sites and O₂ as the terminal oxidant in this catalysis (Table 1, Entries 9 and 10).”

(Page 13, Line 244)

Table R1 | Additional catalysis towards the selective aerobic oxidation of benzyl alcohol (1a) to benzaldehyde (2a) using colloidal gold nanoparticle catalysts.^a

Entry	Catalyst	Yield (%)	Particle size (nm)	Note
1	Au-TOASiW9	75	2.9	–
2	Au-TOAB	3	2.5	–
3 ^b	Au-TOAB-SiW9	48	–	Through ligand exchange from Au-TOAB to TOASiW9
4	Au-dodecanethiol	<1	2.6	–
5 ^c	Au-dodecanethiol + TOASiW9	<1	–	Physical mixture
6 ^d	TOASiW9	<1	–	–
7 ^e	Au-TOASiW9 (Ar 1 atm)	<1	2.9	–

^aReaction conditions: **1a** (0.25 mmol), 3 mL toluene solution of colloidal gold nanoparticles (Au: 4 mol%), K₂CO₃ (0.5 mmol), room temperature (~25 °C), O₂ (1 atm), 24 h. ^bPre-treatment: stirring 2 mL toluene solution of Au-TOAB for 120 min after addition of 1 mL 4 mM **TOASiW9** (toluene). ^cPre-treatment: stirring toluene solution of Au-dodecanethiol for 120 min after addition of 1 mL 4 mM **TOASiW9** (toluene). ^d3 mL toluene solution of **TOASiW9** (4 mM). ^ePre-treatment: freeze-pump-thaw cycles were carried out and the reactor was connected to a balloon filled with an Ar gas. All the conversion and yields were determined via GC analysis using biphenyl as an internal standard.

Attempt in ligand exchange from Au-TOAB to Au-TOASiW9

Fig. R4 | Titration experiments of studying ligand exchange processes through UV-vis characterization about attempts in ligand exchange from Au-TOAB to Au-TOASiW9.

Comment

8. The authors need to provide kinetic data showing catalyst activity over time, as least for the flagship oxidation of benzyl alcohol. Without this, simply showing the particles remain intact by TEM is far from sufficient: it says nothing about activity and stability of the active surface species during catalysis.

Response

According to your comment, the reaction profiles of the aerobic oxidation reaction of benzyl alcohol (**1a**) and 1-phenylethan-1-ol (**1k**) were added (Supplementary Fig. 16). During the reaction, no significant decrease in reactivity was observed for both substrates. These reaction profiles and TEM analysis results showed that colloidal gold nanoparticles maintained stable during catalysis (Supplementary Fig. 16 and 17).

Supplementary Fig. 16 | Reaction profiles of catalytic oxidation of **1a** and **1k**. Reaction conditions: **1a**

or **1k** (0.25 mmol), 3 mL toluene solution of colloidal gold nanoparticles (Au: 4 mol%), Cs₂CO₃ (0.5 mmol), room temperature (~25 °C), O₂ (1 atm), 24 h. All the reaction yields were determined via GC analysis using biphenyl as an internal standard.

Based on above discussions and findings, we added Supplementary Fig. 16 about the reaction profiles, accordingly, we revised and appended the manuscript as follow:

*“Notably, no significant changes were observed in the particle size of the gold nanoparticles protected with **SiW9**, **SiW10** and **SiW11** even after the reaction, and no significant decrease was observed in the catalytic activity of Au-**TOASiW9** during the reaction (Supplementary Figs. 16 and 17).”*

(Page 13, Line 225)

Comment

9. As noted earlier, a full set of analytic data, including ESI-MS and vibrational spectroscopy is needed to prove the POM remains stable during catalysis.

Response

As discussed above, direct observation of structures of POMs is difficult for ESI-MS characterization, thus Raman spectra of Au-**TOASiW9** after the catalytic reaction showed the same characteristic peak to that of Au-**TOASiW9** before the reaction and **TOASiW9**, indicating that POM remains stable during catalysis (Supplementary Fig. 12).

Comment

10. It might be informative to use PW9 to synthesize AuNPs, and determine the catalytic results. Compare the yields with SiW9-stabilized AuNPs to verify the author's hypothesis that AuNP activity is connected to the XPS results. If this were done, 31-P NMR could also be used to characterize the starting materials and the POM-AuNPs colloidal solutions before and after use.

Response

Based on your comment, we prepared Au-**TDASiW9** and Au-**TOAPW9** according to similar procedures with Au-**TOASiW9**. Unlike ultrastable silicotungstate-protected colloidal gold nanoparticles, phosphotungstate-protected gold nanoparticles suffered from instability issue and precipitation of gold was observed within two weeks at room temperature (despite prolonging to one month by storage at refrigerator, Fig. 2b and Fig. R5a). During catalytic oxidation of benzyl alcohol, precipitation of gold colloids was also observed when using Au-**TOAPW9** (Supplementary Fig. 5 and Fig. R5b), and Au-**TOAPW9** exhibited lower activity than that of Au-**TOASiW9** (Table R2). Together with our previous reports that despite immobilization on carbon supports, phosphorous-centered-POM-protected gold nanoparticles possessed significantly low reactivity (*Angew. Chem. Int. Ed.* **2022**, *61*, e202205873), the (electronic) effect of those phosphotungstates may differ from silicon-centered-POMs used in this report, making it difficult to investigate their electronic effect on catalysis and further structural characterization.

a) Photos of gold nanoparticles after storage for two weeks

From left to right:
Au-TDASiW9
Au-TOAPW9

b) Photos of gold nanoparticles after catalysis

From left to right:
Au-TDASiW9
Au-TOAPW9

Fig. R5 | Photographs of different polyoxometalate-protected gold nanoparticles after storage and catalytic oxidation of benzyl alcohol. Reaction conditions: **1a** (0.25 mmol), 3 mL toluene solution of colloidal gold nanoparticles (Au: 4 mol%), Cs₂CO₃ (0.5 mmol), room temperature (~25 °C), O₂ (1 atm), 24 h.

Table R2 | Additional catalysis towards the selective aerobic oxidation of benzyl alcohol (**1a**) to benzaldehyde (**2a**) using colloidal gold nanoparticle catalysts.^a

Entry	Catalysts	Yield (%)	Particle size (nm)	Note
1	Au-TOASiW9	75	2.9	-
2 ^b	Au-TOAPW9	14	2.7	Immediate use

^aReaction conditions: **1a** (0.25 mmol), 3 mL toluene solution of colloidal gold nanoparticles (Au: 4 mol%), K₂CO₃ (0.5 mmol), room temperature (~25 °C), O₂ (1 atm), 24 h. ^bFreshly-prepared colloidal gold nanoparticles were used in catalysis. All the conversion and yields were determined via GC analysis using biphenyl as an internal standard.

Comment

11. If the dehydrogenation mechanism is correct, O₂ should be converted into H₂O during the reaction. ¹⁸O isotope experiments with ¹⁸O₂ should be carried out to confirm this.

Response

Firstly, TOASiW9 did not catalyze alcohol oxidation under current conditions (Table R1, Entry 6), and the reaction did not proceed under argon atmosphere using Au-TOASiW9 (Table R1, Entry 7), indicating that POM ligands did not directly participate in the catalysis. Then, for the alcohol oxidation, O₂ is generally considered to be converted into H₂O during the reaction (*Science* **2010**, *330*, 74–78; *Green Chem.* **2013**, *15*, 17–45; *RSC Adv.* **2016**, *6*, 25279–25285). In addition, owing to the trace amount of produced H₂O during catalysis, hydration with excessive amount of base (K₂CO₃/Cs₂CO₃) also makes it

hard to detect H_2^{18}O after catalysis in the toluene solution. Based on above considerations, we think that $^{18}\text{O}_2$ isotope experiments are not suitable for this system.

Comment

12. The authors proposed a dehydrogenation mechanism in which the first step is the activation of the substrate to generate hydride. If this is the rate limiting step, a significant kinetic isotope effect should be observed using deuterated benzaldehyde as substrate.

Response

Based on your comment, we investigated kinetic isotope effect (KIE) of Au-TOASiW9-catalysed aerobic oxidation of benzyl alcohol and benzyl- $\alpha,\alpha\text{-d}_2$ alcohol under oxygen or air atmosphere respectively (Supplementary Figs. 18 and 19). Under oxygen atmosphere (O_2 , 1 atm), a much higher reaction rate was found for benzyl alcohol as substrate ($k_{\text{H}}/k_{\text{D}} = 3.2$), indicating that C–H cleavage can be the turnover limiting step (TLS); while under air atmosphere (O_2 , 0.2 atm), similar reaction rates for these substrates ($k_{\text{H}}/k_{\text{D}} = 1.2$) suggested that instead of C–H cleavage, oxygen adsorption and/or activation can be TLS.

Considering the higher reaction rate of Au-TOASiW9-catalysed aerobic oxidation of benzyl alcohol under O_2 than air ($k_{\text{O}_2}/k_{\text{Air}} = 2.0$), it is concluded that under air, where O_2 adsorption was TLS, no KIE was observed. In contrast, under O_2 , where O_2 adsorption became saturated and C–H cleavage became TLS, and KIE was observed. Hereafter, besides indirect C–H cleavage pathway by activated oxygen on anionic gold surface proposed in previous reports concerning with polymer/carbon-supported gold nanoparticles (*JACS Au* **2021**, *1*, 660–668; *Angew. Chem. Int. Ed.* **2022**, *61*, e202205873), direct C–H cleavage pathway on gold surface is also possible and actually more general and thus added, while in both pathways, efficient adsorption and/or activation of O_2 is considered essential (Supplementary Fig. 15).

a) Indirect C–H cleavage pathway

b) Direct C–H cleavage pathway

Supplementary Fig. 15 | Possible routes of Au-TOASiW9-catalysed aerobic oxidation of **1a**: **a**, Indirect C–H cleavage pathway. **b**, Direct C–H cleavage pathway. In both possible reaction pathways, efficient activation of O_2 can be considered to facilitate the aerobic oxidation.

Supplementary Fig. 18 | Kinetic isotope effect test for the aerobic oxidation reaction of benzyl alcohol under O_2 (1 atm). Reaction conditions: benzyl alcohol or benzyl- α,α - d_2 alcohol (0.25 mmol), 3 mL toluene solution of Au-TOASiW9 (Au: 4 mol%), biphenyl (0.1 mmol, internal standard), Cs_2CO_3 (0.5 mmol), room temperature ($\sim 25^\circ C$), O_2 (1 atm).

Supplementary Fig. 19 | Kinetic isotope effect test for the aerobic oxidation reaction of benzyl alcohol under air. Reaction conditions: benzyl alcohol or benzyl- α,α - d_2 alcohol (0.25 mmol), 3 mL toluene solution of Au-TOASiW9 (Au: 4 mol%), biphenyl (0.1 mmol, internal standard), Cs_2CO_3 (0.5 mmol), room temperature ($\sim 25^\circ C$), open air.

Based on above discussions and findings, we added another possible direct C–H cleavage pathway in Supplementary Fig. 15 and KIE experiments in Supplementary Figs. 18 and 19, and the revised the manuscript as follows.

“The kinetic isotope effect (KIE) was then examined for Au-TOASiW9-catalysed oxidation of **1a**. Under O_2 atmosphere (1 atm), a much higher reaction rate was observed for **1a** than benzyl- α,α - d_2 alcohol ($k_H/k_D = 3.2$, Supplementary Fig. 18), indicating that C–H cleavage can be the turnover limiting step. When the reaction was carried out under air atmosphere (O_2 , 0.2 atm), no significant KIE was observed ($k_H/k_D = 1.2$, Supplementary Fig. 19), suggesting that O_2 adsorption and/or activation can be turnover limiting step under air atmosphere (Supplementary Fig. 15).”

(Page 12, Line 246)

Comment

13. In arguing the catalyst is stable under turnover conditions, the authors report (Supporting Figure 5) changes in UV-vis spectra after "addition of Cs₂CO₃". No indication is given regarding how much Cs₂CO₃ is added. This is left out entirely. These experiments need to be repeated using the catalytic samples themselves, before and after catalysis. It is very concerning that those obviously necessary data are not included?

Response

Regarding the experiments of UV-Vis spectra in Supplementary Fig. 5, the amount of Cs₂CO₃ was same as the catalytic reactions. As the absorbance peak of benzaldehyde product was located below 360 nm, overlapping absorbance peaks of POMs at 300 nm, we carried out this UV-vis experiment to test stability of colloidal gold nanoparticles towards addition of base without adding substrate.

The experimental details including those additional experiments were added to Method section in the revised manuscript or figure/table captions in the supplementary information as follows:

"UV-vis titration procedure, based on changes in SPR absorbance,^{29,30} was used to quantify the replacement of TOASiW9 by thiolates on the surfaces of the gold nanoparticles. A toluene solution of dodecanethiol (5 mM) was gradually added to a toluene solution of Au-TOASiW9 (0.5 mM Au), and the changes of the absorbance of SPR bands were monitored by UV-vis spectra. The temperature was maintained at 25.0 ± 0.1 °C."

(Page 24, Line 513)

"For the reaction under Ar (1 atm), freeze-pump-thaw cycles were carried out and the reactor was connected to a balloon filled with an Ar gas."

(Page 24, Line 524)

Comment

14. The catalytic results and stability should be carefully compared to those reported by Finke. This kind of colloid was described by Finke et al. as tetraalkylammonium- and POM-co-stabilized metal(0) NPs. The distinctions and similarities should be stated. The authors should also provide evidence that TOA+ is not involved in the stabilization of AuNPs in this work.

Response

Finke reported the preparation of Ir and Rh nanoparticles co-stabilized by TBA cations and metal-substituted POMs in acetone (*J. Am. Chem. Soc.* **1994**, *116*, 8335–8353; *Inorg. Chem.* **1994**, *33*, 4891–4910). Finke's reports and our report both use organic solvents (acetone *versus* toluene) and tetraalkylammonium cations (tetrabutylammonium *versus* tetraoctylammonium) to synthesize POM-protected metal nanoparticles. However, there are some major differences as elaborated hereafter:

i) The most important difference is that we used multivacant lacunary POMs to stabilize metal nanoparticles as well as control the electronic states of the obtained metal particles in non-polar solvents. In addition, our methods can synthesize various POM-protected metal nanoparticles, such as Au, Pt, Re and Rh.

ii) In this work, the most important role of TOA was to endow hydrophobicity of inorganic precursors and formed nanoparticles to retain high stability of colloidal metal nanoparticles in non-polar solvents (e.g. toluene). While on the contrary, by utilization of TBA and CTA, phase transfer of those precursors cannot be achieved (Supplementary Fig. 20). It is clear that TOA hold more bulky structures than TBA, thus according to the called "electrosteric effect" by Finke, TOA-POM-protection should exhibit much better resistance than TBA-POM-protection.

Although TOA can contribute to stabilization of metal nanoparticles, it is noted that the TOA-protected

gold nanoparticles (Au-TOAB) did not hold sufficient stability, and precipitated under the conditions of catalytic reactions (Fig. 2, Supplementary Fig. 4 and 5), revealing the essential roles of multidentate POMs in our method.

iii) Thirdly, from highly negative zeta potential and much lower binding energy of Au_{4f} of Au-**TOASiW9** than Au-TOAB, it is suggested that there exists strong interparticle repulsion by POM protecting layers and robust electronic effect from multi-dentate POM to Au nanoparticles (Fig. 3a, d).

iv) Then, based on theoretical calculations results, drastic decrease of energy was observed by using multi-dentate POM interacting with gold nanoparticles (Fig. 3f), thus relatively weaker interaction of surfactant protection simply through steric effect makes it hard for TOA⁺ to compete in directly interacting with gold nanoparticles than POM protection.

v) Finally, in contrast to the phenomenon that fast agglomeration of gold nanoparticles was observed in acetonitrile during reduction process by Cronin (*Inorg. Chem.* **2019**, *58*, 4110–4116), ultrastable metal nanoparticles were obtained in this method, indicating multi-dentate POM instead of tetraalkylammonium can well protect metal nanoparticles after preventing potential interruption of solvent molecules.

Considering the above factors, it is obvious TOA are involved in the preparation and stabilization of colloidal gold nanoparticles in toluene (and other solvents) solution by electrosteric effect like Finke's work, but they are more essential in achieving non-polar-solvent-based synthesis to solve "semi-stability issue" and more importantly, multi-dentate POM protection is the key to interacting with metal nanoparticles to retain high stability and reactivity in this work.

Comment

The paper reports the synthesis of very stable and highly dispersed colloidal Au particles in polar media using Polyoxometalates (POMs) as stabilising ligands. The particles are stable in the presence of a base, and heating which makes them attractive for oxidation reactions typically carried out on Au. The paper carries out systematic experiments and demonstrates well the performance of these materials

Response

Thank you very much for valuable comments. We have carefully considered all your comments and revised the manuscript. We believe that the addition incorporated in the revised manuscript is appropriate. Please confirm the following responses.

Comment

– however, what is not so clear is the justification of “The robust electronic and moderate steric effects of multi-dentate POM ligands are essential to achieve an extraordinary catalytic performance”. Despite this being a justification for the study (to make stable colloidal particles to study the inherent activity of Au particles), the authors do not separate the catalysts' performance from the stabilizer ligands' role. POMs are known to be highly active in redox especially WO₆ type species reactions – do they contribute to oxygen activation? Are these catalysts more “active” or just more stable so can do more turnovers? Some comparisons to supported samples would help to understand this.

Response

In a recent review article, Weinstock and Neumann summarized catalytic oxidation reactions using O₂ by POMs (*Chem. Rev.* **2018**, *118*, 2680–2717). However, as our experimental results revealed that **TOASiW9** did not catalyze the oxidation of benzyl alcohol under O₂ at room temperature (Table 1, entry 9), and the reaction did not proceed using Au-**TOASiW9** under an Ar atmosphere (Table 1, entry 10), thus indicating it is unlikely that oxygen atoms in POM frameworks participate in the catalytic reaction of this work.

The reasons towards superior reactivity of multidentate POM-protected gold nanoparticles can be summarized as electronic effect from multidentate POM ligands leading to anionic gold surface. As it was elucidated by Tsukuda, Weinstock, and our previous work with efficacy of anionic gold nanoparticles in activating oxygen for catalysis (*J. Am. Chem. Soc.* **2009**, *131*, 7086–7093; *Inorg. Chem.* **2017**, *56*, 2400–2408; *Angew. Chem. Int. Ed.* **2022**, *61*, e202205873), anionic colloidal gold nanoparticles developed in this report exactly exhibit excellent catalytic reactivity. In the two possible reaction pathways (Supplementary Fig. 15), efficient O₂ adsorption and/or activation can be considered as essential steps and thus anionic gold nanoparticles induced by multidentate POM ligands can show high reactivity.

Meanwhile, it is obvious that excellent stability of gold nanoparticles enabled by protection effect of POM ligands is of great significance as well, in comparison to agglomeration and precipitation of gold during catalysis by protection with surfactant TOAB, dodecanethiol and non-vacant POMs (Fig. 2), thus indicating the “stabilizer ligands' role” of POM ligands.

Furthermore, towards the catalytic performance, Au-**TOASiW9**, Au-**TOASiW10** and Au-**TOASiW11** were obtained by using different multidentate POMs ligands and they not only possess similar particle sizes as 3 nm after preparation but also still maintain stable after catalytic reaction (Fig. 2, Supplementary Figs. 6 and 17, Table 1). Based on this, sequentially modulated reactivity was observed in line with electronic states of as-obtained colloidal gold nanoparticles (Fig. 3d, Table 1), thus indicating electronic effect from POMs to gold nanoparticles contributes to facilitating the reaction instead of just stabilizing gold nanoparticles.

In summary, multidentate POM ligands are employed in stabilizing gold nanoparticles under different reaction conditions, and simultaneously have the electronic effect on gold nanoparticles contributing to the efficient O₂ adsorption/activation and thus catalysis.

Comment

In addition, the authors correlate the performance to the nature of the POM suggesting that the POMs which are negatively charged bind to the Au particles and generate negative Au surfaces (via XPS binding energy correlations) – in this case what are the counterions to the Au surfaces? Do the authors suggest electron transport from the POM to the Au?

Response

To fulfill charge balance, it is considered that TOA acted as counter cations for the POM-protected gold nanoparticles. Considering the bulky structures of TOA cations and POMs, and stronger interaction of POM rather than TOA to metal surface results in possible structures, we think that POMs and TOA formed inner and outer layers of gold nanoparticles, respectively (Supplementary Fig. 9d). This structure hypothesis can be confirmed by i) drastic decrease of energy of multi-dentate POM interacting to gold nanoparticles comparing to simply steric effect of surfactant protection (Fig. 3g); ii) highly negative zeta-potential of Au-TOASiW9 and almost neutral zeta-potential of Au-TOAB suggesting that POMs are located near Au surface (Fig. 3a); iii) electronic effect from multi-dentate POMs to gold nanoparticles indicated by significantly changed binding energy of Au_{4f} orbitals (Fig. 3d); and iv) previous reports of Weinstock in directly observing POM ligands surrounding metal nanoparticles (*J. Am. Chem. Soc.* **2008**, *130*, 16480–16481; *J. Am. Chem. Soc.* **2009**, *131*, 17412–17422).

The electronic effect of POM ligands to gold nanoparticles was supported by the results of XPS characterization and catalytic reactions (Fig. 3d, Table 1). Weinstock and Wang first proposed that anionic POM ligands can have adjustable electronic effects to gold surface by POMs with different negative charges and applied them into CO oxidation (*Inorg. Chem.* **2017**, *56*, 2400–2408). This phenomenon of “electron transport from the POM to the Au” has been utilized by Newton et al. and Yamazoe et al., respectively, to confirm the modification effect of POM ligands for gold nanoparticles since then (*Angew. Chem. Int. Ed.* **2020**, *59*, 14331–14335; *Chem. Commun.* **2022**, *58*, 9018–9021). In our recent report, we further confirmed that multiple vacant sites of POM ligands can be the key factor in strengthening the electronic effect, and created supported anionic gold nanoparticles for the first time (*Angew. Chem. Int. Ed.* **2022**, *61*, e202205873). Here in this work, changes in the binding energy of gold nanoparticles (Fig. 3d) together with the results of catalytic reactions (Table 1) can confirm the electronic effect of POMs to Au nanoparticles.

REVIEWER COMMENTS

Reviewer #1 (Remarks to the Author):

The authors have revised their manuscript carefully based on the referee comments provided. I have gone through all reviewer comments and author responses, and I believe the authors have addressed the comments. With respect to the novelty of the study compared to the literature-reported studies, and the impact for publication in Nature Communications, I believe that the general approach, wide application scope and detailed mechanistic picture provided in the revised manuscript warrant publication of the manuscript in Nature Communications.

Reviewer #2 (Remarks to the Author):

In their revised manuscript the authors provide a significant amount of newly added data which provide support for a number of centrally important claims that in the original manuscript were less adequately defended. For example, the authors now provide considerably more characterization data, including UV-Visible documentation of the displacement of POM ligands from the Au NP surface upon titration by alkanethiol, transfer of POM ligands into water followed by spectroscopic characterization, additional control experiments demonstrating a clear role for POM ligands at the Au surface in catalysis and kinetic isotope effect data in support of a proposed mechanism. Additional experimental conditions have been clarified, or made more readily accessible to the reader and scholarly context of this work has been improved by a more inclusive discussion and citations to primary literature. In some cases, where requested data have not been provided, the authors provide reasonable justifications, largely based on understandable practical issues. Having provided the above new information, the authors' argument that the trilacunary "SiW9" cluster anion is indeed intact and associated with the Au NP surface under their new synthetic conditions and after some use in catalysis, and that the new organic-soluble catalyst is capable of carrying out aerobic transformations that represent a significant advance with clear options for future developments based on reductions of metal salts in organic solvents in the presence of organic-solvent soluble forms of multi-vacant POM cluster anions.

Before publication, the authors should repeat or at least provide a defensible explanation for the results of one newly added experiment that, while superficially appearing to provide a definitive answer, appears to not make sense in its details. This refers to Figure R4 in which tetraoctylammonium (TOA) bromide protected Au NPs are titrated by TOASiW9. Incremental additions of the POM salt cause a decrease in adsorbance at the SPR maximum.

In Supplementary Figure 9, the same band decreases in adsorbance upon replacement of POM at the Au surface by thiolate ligands. This is because the refractive index of the organic ligands is smaller than that of the POM. As such, the opposite behaviour should be observed in Figure R4, where POM is replacing organic ligands at the Au surface. That is, adsorbance at the SPR maximum should increase. The decrease in Figure R4 does not make sense.

Furthermore, there is no reason to expect a breakpoint during POM addition. Rather there should be a gradual approach to a maximum adsorbance value. This is necessarily the case where the POM possesses some lability (as the authors claim to be the case) and as a result, increase in POM concentration leads asymptotically to a limiting value. Why do the authors observe a breakpoint for POM addition? This is inconsistent with their understanding of the lability of the POM-Au interaction.

Finally, returning briefly to the issue of scholarly context, the authors provide in their introductory graphic (Figure 1a) a summary of ligand protected Au NPs. They include thiolate, other organic polymers and POMs. For the POM case, they refer only to plenary POMs, leaving out considerable work by Weinstock and several others, in which mono-vacant POMs form monolayers at metal surfaces

giving indefinitely stable and in some cases catalytically active solutions in water. The graphic as written implies by omission that no other POM protected Au NPs display the stability of the ones reported here. This diminishes the scholarly integrity of the manuscript.

Reviewer #3 (Remarks to the Author):

The authors have clearly answered the comments i made previously. In addition is is clear that they have taken great efforts to answer the points of the other reviewers.

I have no further comments.

<As for Reviewer 1>

Comment

The authors have revised their manuscript carefully based on the referee comments provided.

I have gone through all reviewer comments and author responses, and I believe the authors have addressed the comments. With respect to the novelty of the study compared to the literature-reported studies, and the impact for publication in Nature Communications, I believe that the general approach, wide application scope and detailed mechanistic picture provided in the revised manuscript warrant publication of the manuscript in Nature Communications.

Response

Thank you very much for high evaluation of our manuscript.

<As for Reviewer 2>

Comment

In their revised manuscript the authors provide a significant amount of newly added data which provide support for a number of centrally important claims that in the original manuscript were less adequately defended. For example, the authors now provide considerably more characterization data, including UV-Visible documentation of the displacement of POM ligands from the Au NP surface upon titration by alkanethiol, transfer of POM ligands into water followed by spectroscopic characterization, additional control experiments demonstrating a clear role for POM ligands at the Au surface in catalysis and kinetic isotope effect data in support of a proposed mechanism. Additional experimental conditions have been clarified, or made more readily accessible to the reader and scholarly context of this work has been improved by a more inclusive discussion and citations to primary literature. In some cases, where requested data have not been provided, the authors provide reasonable justifications, largely based on understandable practical issues. Having provided the above new information, the authors' argument that the trilacunary "SiW₉" cluster anion is indeed intact and associated with the Au NP surface under their new synthetic conditions and after some use in catalysis, and that the new organic-soluble catalyst is capable of carrying out aerobic transformations that represent a significant advance with clear options for future developments based on reductions of metal salts in organic solvents in the presence of organic-solvent soluble forms of multi-vacant POM cluster anions.

Response

Thank you for your important comments on our revised manuscript. We have carefully considered all your comments. We believe that the following point-to-point responses to your comments and the revised Figure 1 are appropriate.

Comment

Before publication, the authors should repeat or at least provide a defensible explanation for the results of one newly added experiment that, while superficially appearing to provide a definitive answer, appears to not make sense in its details. This refers to Figure R4 in which tetraoctylammonium (TOA) bromide protected Au NPs are titrated by TOASiW₉. Incremental additions of the POM salt cause a decrease in adsorbance at the SPR maximum.

In Supplementary Figure 9, the same band decreases in adsorbance upon replacement of POM at the Au surface by thiolate ligands. This is because the refractive index of the organic ligands is smaller than that of the POM. As such, the opposite behaviour should be observed in Figure R4, where POM is replacing organic ligands at the Au surface. That is, absorbance at the SPR maximum should increase. The decrease in Figure R4 does not make sense.

Furthermore, there is no reason to expect a breakpoint during POM addition. Rather there should be a gradual approach to a maximum absorbance value. This is necessarily the case where the POM possesses some lability (as the authors claim to be the case) and as a result, increase in POM concentration leads asymptotically to a limiting value. Why do the authors observe a breakpoint for POM addition? This is inconsistent with their understanding of the lability of the POM-Au interaction.

Response

We strongly agree that the ligand exchange method by Weinstock, Wang and co-workers from citrate to POM ligands is a big success in catalytic study in water (*Inorg. Chem.* **2017**, *56*, 2400–2408). In the reports by Weinstock and co-workers, it has been reported that the absorbance of SPR band of citrate-protected Au nanoparticles increased and the band was shifted to longer wavelength by ligand exchange with POMs in water. Whereas, as discussed in our previous point-by-point responses to your comments, the ligand exchange experiment for Au-TOAB with **SiW9** (Figure R4) showed i) the maintained SPR wavelength, ii) gradually elevated absorbance at 700 nm, and iii) obviously lower catalytic activity of Au nanoparticles obtained through ligand exchange from Au-TOAB than that of Au-**TOASiW9**. We considered that these results suggested that agglomeration of some gold nanoparticles occurred during the ligand exchange process and possessed different structures from directly prepared Au-**TOASiW9**. As you pointed out, the usage of “breakpoint” was deleted in Fig. R4 in order to avoid possible misunderstandings.

Fig. R4 | Titration experiments of studying ligand exchange processes through UV-vis characterization about attempts in ligand exchange from Au-TOAB to Au-**TOASiW9**.

Comment

Finally, returning briefly to the issue of scholarly context, the authors provide in their introductory graphic (Figure 1a) a summary of ligand protected Au NPs. They include thiolate, other organic polymers and POMs. For the POM case, they refer only to plenary POMs, leaving out considerable work by Weinstock and several others, in which mono-vacant POMs form monolayers at metal surfaces giving indefinitely stable and in some cases catalytically active solutions in water. The graphic as written implies by omission that no other POM protected Au NPs display the stability of the ones reported here. This diminishes the scholarly integrity of the manuscript.

Response

Thank you for your comment. According to your comment, we revised Figure 1a by adding depicts of both plenary and lacunary POMs and listed mono-vacant $[\text{AlW}_{11}\text{O}_{39}]^{9-}$ and tri-vacant $[\text{SiW}_9\text{O}_{34}]^{10-}$ as representative POM ligands. Also, considering that there are in some cases where “agglomeration during practical use” can be avoided to some extent, this description was deleted from Figure 1.

a Representative protection methods for gold nanoparticles

b This work: a non-polar-solvent-based multi-dentate polyoxometalate protection method

- ✓ A universal protocol for achieving gold nanoparticles with high stability, activity and selectivity
- ✓ Wide applicability to various polyoxometalates, surfactants, solvents and metals

Fig. 1 | Preparation of gold nanoparticles. **a**, Representative methods for the preparation of gold nanoparticles using thiol protection, organic polymer protection and polyoxometalate protection. **b**, This work: a non-polar-solvent-based multi-dentate polyoxometalate protection method for developing ultra-stable and highly reactive gold nanoparticle catalysts.

<As for Reviewer 3>

Comment

The authors have clearly answered the comments i made previously. In addition is is clear that they have taken great efforts to answer the points of the other reviewers.

I have no further comments.

Response

Thank you very much for high evaluation of our manuscript.